# Efficiently Enhancing Zero-Shot Performance of Instruction Following Model via Retrieval of Soft Prompt

**Seonghyeon Ye     Joel Jang     Doyoung Kim     Yongrae Jo     Minjoon Seo**
KAIST
{seonghyeon.ye,joeljang,ikevin98,yongrae,minjoon}@kaist.ac.kr

## Abstract

Enhancing the zero-shot performance of instruction-following models requires heavy computation, either by scaling the total number of training datasets or the model size. In this work, we explore how retrieval of soft prompts obtained through prompt tuning can *efficiently* assist hard prompts in zero-shot task generalization. Specifically, we train soft prompt embeddings for each prompt through prompt tuning, store the samples of the training instances mapped with the prompt embeddings, and retrieve the corresponding prompt embedding of the training instance closest to the query instance during inference. While only adding 0.007% additional parameters, retrieval of soft prompt enhances the performance of T0 on unseen tasks by outperforming it on 10 out of 11 datasets as well as improving the mean accuracy of T0 on BIG-bench benchmark by 2.39% points. Also, we report an interesting finding that retrieving source embeddings trained on similar *answer choice formats* is more important than those on similar task types.[1]

## 1 Introduction

Training Large Language Models (LLMs) on huge amounts of data has enabled LMs to perform downstream tasks without any fine-tuning with the aid of natural prompts or concatenation of a few demonstration instances (Brown et al., 2020; Rae et al., 2021; Kojima et al., 2022; Chowdhery et al., 2022). Additionally, recent works have shown that adding a *instruction tuning* stage, an additional training step that helps pretrained LMs understand prompts and demonstrations results in a significant performance boost on zero-shot task generalization even for moderate-sized LMs (Min et al., 2021; Sanh et al., 2021; Wei et al., 2021; Wang et al., 2022b; Ye et al., 2022; Chung et al., 2022). This extra

---

[1]Model checkpoints and code implementation are available at github.com/seonghyeonye/RoSPr.

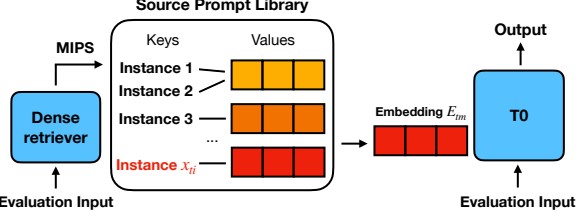

Figure 1: During zero-shot inference, RoSPr selects similar training instances with the given input from the *Source Prompt Library* and retrieves the prompt embeddings corresponding to the selected training instances.

instruction-tuning stage involves explicit, multi-task prompted learning on various tasks, enabling LMs to quickly adapt to unseen tasks at inference.

To maximize the effect of instruction-tuning, two approaches have been widely explored: (1) scaling the number of training datasets, and (2) scaling the model size (Wang et al., 2022b; Chung et al., 2022). However, both approaches require heavy computation, not applicable with an academic budget. Specifically, the first approach requires updating the whole parameters of the model every time a training dataset is added, showing limitations in terms of scalability. On the other hand, the second approach requires heavy memory requirements to load and train a massive LLM.

To enhance the zero-shot performance of instruction-following model efficiently, we introduce **R**etrieval **o**f **S**oft **Pr**ompt (RoSPr), which is easily scalable and requires minimal computation by only adding 0.007% parameters to the main model during inference. As shown in Figure 1, by training prompt embeddings (soft prompt) for each given hard prompt through prompt tuning, we construct a *Source Prompt Library* consisting of samples of training instances mapped with their corresponding prompt embeddings. Then, during inference, by using a simple, off-the-shelf dense retriever model, we search for training instances similar to the given query instances and retrieve their corresponding prompt embeddings. Because the backbone LM is frozen, it allows the retrieved

embeddings to serve as adapters assisting hard prompts. While ROSPR can be applied to any LM, in this work, we use T0 (Sanh et al., 2021) as our initial backbone LM and perform prompt tuning on the tasks used during the instruction-tuning stage.

While adding only 0.007% additional parameters, ROSPR outperforms T0 on 10 out of 11 evaluation datasets and outperforms efficient fine-tuning baselines without any target task fine-tuning. ROSPR is also effective for challenging tasks such as tasks from BIG-bench (Srivastava et al., 2022), outperforming T0 by 2.39% mean accuracy. Furthermore, we provide several interesting findings: (1) Variants of ROSPR that include interpolation of multiple prompt embeddings and scoring method that considers the answer choice distribution during retrieval further increases the effect of ROSPR (2) Also, we provide analysis of which factors attribute to the performance of ROSPR and show that, similarly to the role of demonstrations in in-context learning (Min et al., 2022), heuristic features such as *answer choice format* are more important than the similarity of the source task.

## 2 Related Work

### 2.1 Task Generalization with Instruction-Tuning

Prompts and demonstrations are essential for task generalization since proper explanations are required for LMs to understand an unseen task (Kojima et al., 2022; Wei et al., 2022; Lampinen et al., 2022). *Instruction-tuning*, which is *explicit* multi-task prompted training on various downstream tasks, is a simple but effective way to achieve this, resulting in improved zero-shot capabilities. Zhong et al. (2021) first introduced the method of instruction-tuning by converting various tasks into a question-answering format and finetuning the model on the aggregated dataset. Following works (Mishra et al., 2022; Min et al., 2021; Sanh et al., 2021; Wei et al., 2021; Wang et al., 2022b; Xu et al., 2022; Ouyang et al., 2022; Ye et al., 2022; Chung et al., 2022) extended this approach on a larger scale and show that zero-shot task generalization could be enhanced with more diverse prompts, a larger number of training downstream tasks, and a larger LM.

### 2.2 Source Task Retrieval

Retrieving a source task that is relevant to the target task has shown to result in faster and better task

adaptation. For parameter-efficient fine-tuning, Vu et al. (2022); Su et al. (2022) retrieve source prompt embedding that is similar to the target prompt embedding and obtain a better initialization point for prompt tuning. Instead of utilizing a single prompt embedding, recent works show a mixture of multiple prompt embeddings to be effective (Asai et al., 2022; Qin and Eisner, 2021).

For instruction-tuning, Lin et al. (2022) retrieve training instances that are similar to the query through a dense retriever and fine-tune the model using the retrieved examples. For in-context learning, Rubin et al. (2021); Liu et al. (2022b); Wang et al. (2023) retrieve training data that could be used for demonstrations. Wang et al. (2022c) show the effect of retrieving prompt embeddings in a continual learning setting. Although our proposed method is related to these works, the novelty of our work lies in applying source task retrieval in the zero-shot setting and retrieving soft prompts instead of training instances.

## 3 Method

In this section, we introduce Retrieval of Prompt Tuning (ROSPR) for zero-shot task generalization. A detailed overview is shown in Figure 2. We first train source prompt embeddings of LM for each *hard* prompt given a source task using prompt tuning (Section 3.1). Then, we save training instance samples along with their prompt embeddings in the *Source Prompt Library* and use it to retrieve embeddings at inference to perform tasks in a zero-shot manner (Section 3.2). We additionally introduce interpolation of multiple source prompt embeddings (Section 3.3) and variance-based ranking (Section 3.4) to increase robustness and accuracy.

### 3.1 Training Source Prompt Embeddings

Even though ROSPR may be used to augment any type of LM, we use T0 (Sanh et al., 2021) as the backbone LM for this paper. For training of *soft* prompts, we utilize the source tasks and prompts used for the instruction-tuning phase of T0. While T0 was trained in a multi-task learning manner, we freeze the initial T0 parameters and train only *soft* prompts (source prompt embeddings) for each hard prompt of the source task.

**Prompt Tuning** Among various parameter-efficient fine-tuning methods, we follow prompt tuning proposed by Lester et al. (2021) because the number of trainable parameters is extremely small

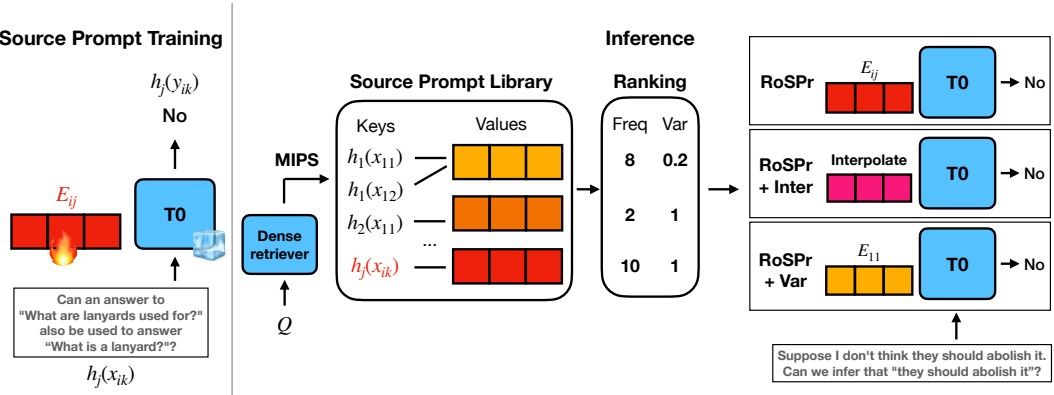

Figure 2: An overview of RoSPr. For each hard prompt of the source datasets, soft prompts are trained via prompt tuning. After storing training instances as keys and corresponding prompt embedding as values, RoSPR searches training instances similar to query set $Q$, retrieves the corresponding prompt embeddings, and selects the most frequently retrieved candidate for inference. Variants of selection strategy are also shown: RoSPR+INTER interpolates between multiple related source embeddings and RoSPR+VAR ranks candidate embeddings considering both frequency and variance.

($\sim$204K parameters per prompt), which implies that the memory overhead of parameter retrieval at inference is negligible.

For each source training dataset $D_i$ ($i = 1, .., T$) where $T$ is the total number of source datasets, we train source embeddings $E_{ij}$ ($j = 1, .., M_i$) where $M_i$ is the number of hard prompts in $D_i$, making soft prompt embeddings for each individual hard prompts. Specifically, given a training instance $\{x_{ik}, y_{ik}\}$ ($k = 1, .., K$) from $D_i$ where $K$ is the number of sampled training instances per dataset, we first convert it into its *hard* prompted version $\{h_j(x_{ik}), h_j(y_{ik})\}$ where $h_j(\cdot)$ denotes adding the $j$-th *hard* prompt [2]. Next, we train the LM with the following objective:

$$\max_{E_{ij}} P(h_j(y_{ik})|E_{ij}; h_j(x_{ik})) \qquad (1)$$

where all the parameters of the underlying backbone LM are frozen and only $E_{ij}$ is trainable. In short, given $D_i$, we perform $M_i$ number of prompt tunings for each *hard* prompts, resulting in $\sum_{i=1}^{T} M_i$ total number of source prompt embeddings. For training efficiency, we only train on $K = 5000$ training instances for a single epoch for each source prompt embedding.

### 3.2 Zero-Shot Embedding Retrieval

After source prompt embedding training, we retrieve the most related source embeddings and select one from the retrieved candidates to be used at inference (right part of Figure 2).

We first construct a *Source Prompt Library*, consisting of sentence-level representations of training instance inputs as keys and the corresponding source prompt embedding as the values. For each available source prompt embedding, *n* number of samples are stored in the library. The sentence-level representations are obtained by getting the mean representation of hidden states of the last layer of the dense retriever. We use a T0-small encoder as a dense retriever, replicated based on Sanh et al. (2021) with smaller model size.

At inference, we first randomly sample $Q$ query instances from the target task, following Lin et al. (2022). After obtaining sentence-level representations for each query through our T0-small encoder, we retrieve top-$N$ examples for each query instance using MIPS (maximum inner product search) operation [3] on our Source Prompt Library, retrieving a total of $Q * N$ prompt embeddings. As the default methodology, among the retrieved embedding candidates, we select the most frequently retrieved prompt embedding as our designated *soft* prompt for the given target task and concatenate the embedding with each of the target task instances before feeding it to our backbone LM. In the next two subsections, we explain different strategies for calculating the target embedding from the $Q * N$ prompt embedding candidates.

### 3.3 Interpolation of Prompt Embeddings

When retrieving only a *single* prompt embedding for a given task (Section 3.2), it may result in high variance across evaluation prompts when the selected prompt embedding does not fit well with the

---

[2]For each instances, the input and output are converted into its *prompted* version using the promptsource toolkit (Bach et al., 2022). An example is given in Appendix G.

[3]For all indexing and searching, we use FAISS (Johnson et al., 2019) for fast source prompt embedding retrieval.

given task. Recent works on prompt embedding retrieval have shown that the interpolation of prompt embeddings effectively transfers to the target task (Asai et al., 2022; Vu et al., 2022). We also explore calculating the target embedding through interpolation of multiple source embeddings instead of just using a single embedding. Among $Q * N$ prompt candidates searched in Section 3.2, we select top-$N'$ candidate embeddings based on the frequency of the search. Then, we calculate the weighted sum of the candidate embeddings, where the interpolation weight for each source embedding is based on the proportion of frequency. While Asai et al. (2022); Vu et al. (2022) require fine-tuning the target embeddings on the target task to calculate the weights for interpolation, our approach does not require any target task fine-tuning, enabling zero-shot task transfer.

### 3.4 Variance-based Ranking

Similar to the scoring and calibration method of Lu et al. (2022); Zhao et al. (2021), we introduce a scoring method applicable to zero-shot classification tasks that allows ranking the $Q * N$ retrieved prompt embedding candidates by considering the *answer choice distribution* of the given target task as extra cues together with the original *frequency* cues. To accomplish this, we perform a forward pass with the concatenation of each candidate prompt embeddings together with the given *hard* prompt (only including the instruction, excluding the input instance) of the target task and give a higher score to the embedding candidate that results in lower *variance*. Ideally, the combination of soft and hard prompts should result in equal probability among the answer choices because the actual context of the task is not included.

Specifically, when given a target task with $k$-th hard prompt $h_k$, for each candidate embedding $E_{ij}$, we calculate the scoring as follows:

$$\text{Score}(h_k, E_{ij}) = \frac{\text{freq}(h_k, E_{ij})}{\sqrt{\text{Var}[P(y|E_{ij}, h_k)]}} \quad (2)$$

where $y$ refers to the available output options of the target task.

## 4 Experimental Settings

In this section, we explain the experimental settings of training of source prompt embedding and construction of our *Source Prompt Library*. We also explain our evaluation setting during zero-shot inference and baseline models. We provide detailed experiment configurations in Appendix F.

### 4.1 Source Tasks

For training soft prompts through prompt tuning, we use the subset of source tasks used for the initial T0 instruction-tuning (Sanh et al., 2021) [4]. For each source task, we use the prompts for each dataset in T0, resulting in a total of 230 prompts. For Source Prompt Library construction, we sample only $n = 100$ training instances per source embedding to minimize the inference latency. We show a variation of $n$ and different methods to sample $n$ training instances in Appendix D.

### 4.2 Evaluation Tasks

Following Sanh et al. (2021), we evaluate on the validation set of 4 held-out tasks (natural language inference, sentence completion, coreference resolution, word sense disambiguation) resulting in a total of 11 evaluation datasets. We also follow Sanh et al. (2021) and evaluate on 14 different datasets from the BIG-bench benchmark (Srivastava et al., 2022) [5]. We use rank classification evaluation method by selecting the output option with higher log-likelihood following Brown et al. (2020); Sanh et al. (2021). For all evaluation tasks, we use accuracy as an evaluation metric and report the mean accuracy and standard deviation of all of the evaluation prompts per given dataset (average of ~10 prompts per evaluation dataset) [6]. For BIG-bench tasks, we do not report standard deviation because only one prompt is provided per task.

### 4.3 Baseline Models

**Zero-shot Baseline** For zero-shot baseline models, we show the results of T0 (3B) together with a 4 times larger T0 (11B) instruction-tuned model. We also compare with GPT-3 (175B) model which is 60 times larger than T0 (3B).

**Fine-tuning Baseline** We also compare with efficient fine-tuning baseline models that utilize prompt tuning. These models require target task prompt tuning, indicating that zero-shot transfer

---

[4]We use 29 out of 38 datasets that are used to train T0. We explain the training task selection rationale in Appendix H.1

[5]We provide the full list of evaluation datasets in Appendix H.

[6]For methods based on ROSPR, we report the performance average of 3 runs with different random seeds for the sampling of evaluation queries used for the prompt retrieval.

| Method | # of Param (Base/Trainable) | NLI | | | | | Sentence Completion | | | Coreference Resolut. | | WSD | Total Avg. | |
|---|---|---|---|---|---|---|---|---|---|---|---|---|---|---|
| | | RTE | CB | AN. R1 | AN. R2 | AN. R3 | COPA | Hellasw. | StoryC. | Winogr. | WSC | WiC | *Mean* | *STD* |
| T0 | 3B / 0 | 64.55 | 45.36 | 33.81 | 33.11 | 33.33 | 75.88 | 26.60 | 84.03 | 50.97 | **65.10** | 50.69 | 51.22 | 3.62 |
| PT (FT) | 3B / 204K | 63.14 | 44.52 | 33.07 | 31.44 | 32.94 | 73.00 | 26.39 | - | 50.67 | 46.73 | 50.02 | - | - |
| ATTEMPT (FT) | 3B / 614K | 68.95 | 44.88 | 36.19 | **34.73** | **34.81** | 74.38 | 26.76 | - | 51.33 | 64.90 | 51.05 | - | - |
| T0+ROSPR | 3B / 204K | 71.54 | 49.48 | 37.05 | 34.64 | 33.92 | 78.75 | 26.97 | 85.52 | **51.50** | 64.52 | 51.76 | 53.24 | 3.62 |
| W/ INTER | 3B / 204K | 70.71 | **52.30** | **37.30** | 34.34 | 33.89 | 78.25 | 26.94 | **85.62** | 51.10 | 64.52 | 50.73 | 53.24 | **3.30** |
| W/ VAR | 3B / 204K | 71.78 | 50.36 | 37.07 | 34.58 | 33.90 | **78.88** | **27.01** | 85.52 | 51.45 | 64.94 | **51.94** | 53.38 | 3.38 |
| W/ VAR & INTER | 3B / 204K | **72.60** | 51.98 | 37.25 | 34.31 | 33.95 | 77.83 | 26.84 | 85.58 | 50.93 | 64.97 | 51.18 | **53.40** | 3.47 |
| ORACLE | 3B / 204K | 73.79 | 58.10 | 37.65 | 34.92 | 34.91 | 81.13 | 27.75 | 87.57 | 52.36 | 68.17 | 55.26 | 55.60 | 3.07 |
| T0 | 11B / 0 | 80.83 | 70.12 | 43.56 | 38.68 | 41.26 | 90.02 | 33.58 | 92.40 | 59.94 | 61.45 | 56.58 | 60.76 | |
| GPT-3 | 175B / 0 | 63.50 | 46.40 | 34.60 | 35.40 | 34.50 | 91.00 | 78.90 | 83.20 | 70.20 | 65.40 | 45.92 | 59.00 | - |

Table 1: ROSPR refers to our main proposed method, W/ INTER refers to applying interpolation of multiple source embedding candidates, W/ VAR refers to retrieval through variance-based ranking, W/ VAR & INTER refers to applying both interpolation and variance-based ranking where the interpolation weight is based on the variance-based ranking score, and ORACLE refers performance when the most optimal source embedding is retrieved from the candidates, acting as an upper bound performance for retrieval. FT refers to fine-tuned models on the target tasks. For FT models, we exclude StoryCloze due to the absence of training instances. The best and second-best performance is shown in **bold** and underline respectively. Comparison with hard prompt optimization techniques and visualization of the results is shown in Appendix A and Appendix E, respectively.

is infeasible. Similar to our source prompt tuning process, we train each target prompt for a single epoch with a maximum of 5,000 training instances. The first baseline model is naive prompt tuning on the target tasks without any prompt retrieval, referred to as PT (Lester et al., 2021). The second model is ATTEMPT (Asai et al., 2022), which trains the target soft prompts through attentional mixtures of source prompts. Because StoryCloze (Mostafazadeh et al., 2016) does not contain training instances, we exclude the dataset for fine-tuning. More training details of fine-tuning baseline are specified in Appendix C.

## 5 Experimental Results

**ROSPR enhances the performance of T0 efficiently.** Table 1 shows the performance of the 11 evaluation datasets. T0+ROSPR outperforms T0 on 10 datasets among the 11 evaluation datasets. Specifically, T0+ROSPR outperforms T0 on RTE (+6.99% points), CB (+4.12% points), ANLI R1 (+3.24% points), and COPA (+2.87% points). This shows that soft prompt retrieval assists hard prompts for zero-shot task generalization with a negligible number of additional parameters (0.007%) [7]. Also, while T0 outperforms GPT-3 on 3 datasets (RTE, StoryCloze, WiC), T0+ROSPR additionally outperforms GPT-3 on 2 datasets (ANLI R1 and CB) and enlarging the score gap for RTE, StoryCloze and WiC.

ROSPR also outperforms finetuning baselines even without utilizing any training instances of the target task. We first observe that PT harms the performance of the backbone model, which aligns with the result of Liu et al. (2022a); Gu et al. (2022) that prompt tuning is unstable when the training instances or the training steps are small. By comparing ATTEMPT with ROSPR, ROSPR outperforms ATTEMPT on 7 out of 10 tasks and 1.21% points on the mean accuracy of 10 tasks. This shows that ROSPR is more applicable for efficient adaptation because it requires 3 times fewer additional parameters compared to ATTEMPT and also does not require any further fine-tuning of the target task.

**INTER and VAR enhance the performance of ROSPR.** We also analyze the effect of introducing variants of ROSPR: interpolation of soft prompts (INTER) and variance-based ranking (VAR) in Table 1. First, applying INTER shows similar accuracy compared to ROSPR. However, as shown in the last column of Table 1, INTER reduces the standard deviation of T0 and ROSPR by 8.84% while improving the mean accuracy of T0, indicating increased robustness to different surface forms of evaluation prompts. This indicates that interpolation of multiple source embeddings outperforms a single source embedding retrieval, aligning with the result of Asai et al. (2022). Applying VAR with T0+ROSPR improves both zero-shot accuracy and robustness of T0+ROSPR, showing that considering the answer choice distribution is beneficial for zero-shot setting, aligned with results from Zhao et al. (2021); Shi et al. (2022). Moreover, applying both VAR+INTER results in the highest overall average accuracy, outperforming T0 by 2.18% points by largely reducing the gap between larger LLMs.

---

[7] One exception is WSC, which is a binary classification task (yes/no) predicting whether the reference of the pronoun is correct. We observed that the evaluation data of this dataset has unbalanced labels, containing over 60% of "No" labels. This might be the reason why T0-11B underperforms T0-3B only on this dataset (Sanh et al., 2021). Indeed, predicting only "No" on this dataset outperforms T0-11B (63.46>61.45).

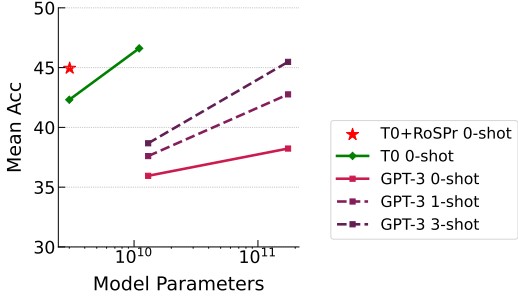

Figure 3: Mean accuracy of 14 datasets of BIG-bench. We evaluate on a single prompt following Sanh et al. (2021). By only adding 0.007% parameters to T0-3B, T0+RoSPR largely reduces the performance gap between 4 times larger T0-11B. The full result is provided in Appendix B.

**The effect of RoSPR generalizes to challenging tasks.** RoSPR is also effective for challenging tasks such as tasks from BIG-bench benchmark. As shown in Figure 3, T0+RoSPR improves the mean accuracy performance of T0-3B by 2.39% points while only adding 0.007% additional parameters. T0+RoSPR also outperforms 60 times larger zero-shot and 1-shot GPT-3 and largely reduces the performance gap between 4 times larger T0-11B (1.84% points) or 60 times larger 3-shot GPT-3 (0.53% points). Applying INTER with T0+RoSPR results in additional mean accuracy enhancement, outperforming T0-3B by 2.67% points [8].

## 6 Analysis of RoSPR

Zero-shot task adaptation of LMs is often seen as a problem of *task location*, locating the target task to where the model can solve it using the intrinsic ability obtained at pretraining stage with the aid of prompts and demonstrations (Reynolds and McDonell, 2021). In this section, we analyze which factors contribute to the performance enhancement in the perspective of identifying better *task location*. We find that although the target task performance depends on the source task types, heuristic features such as the *answer choice format* are more important. This agrees with previous findings that a meta-trained LM focuses on simple features such as the label space, the input distribution, and sequence format, instead of complex semantics (Webson and Pavlick, 2021; Min et al., 2022).

**Target task performance depends on source task types.** To analyze the effect of different source task types on each target task, we measure the frequency ratio of each source task type that results

---

[8]We observe that applying VAR results in the same performance as T0+RoSPR because frequency of retrieval has much more influence than variance for BIG-bench tasks.

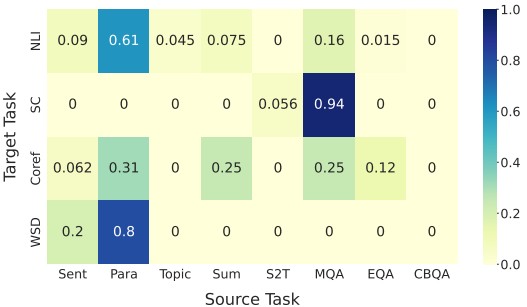

Figure 4: Frequency of source task types (x-axis) that maximizes (i.e. ORACLE) the accuracy of each target task (y-axis).

in the best performance (ORACLE) for the given prompts of the target tasks (visualized in Figure 4). From this figure, we can observe a few patterns: paraphrase task assists NLI and word sense disambiguation task while multi-choice QA (MQA) task assists sentence completion task. For coreference resolution task, various source task types (paraphrase, summarization, multi-choice QA) assist the target task.

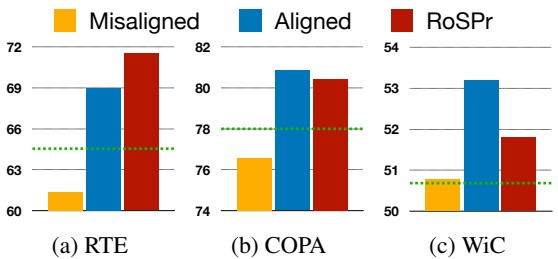

Figure 5: Effect of answer choice format alignment across different target datasets (RTE, COPA, WiC). We report the mean accuracy of the evaluation prompts and the performance of T0 is shown in green dotted line.

**Answer choice format is important for task location.** We also analyze the effect of using different answer choice formats with the same source task. *Answer choice format* decides how the available answer choices are given to the LM through the input. For example, a prompt that requires classifying a movie review into good/bad has a different answer choice format from classifying it into positive/negative.

We experiment on 3 datasets (RTE, COPA, WiC) which correspond to different tasks (NLI, sentence completion, word sense disambiguation) respectively. For each dataset, we select a source dataset that is retrieved the most for ORACLE. Among the source prompts of the selected source dataset, we select a prompt that has the same answer choice format as the target task (ALIGNED) and another prompt that has a different answer choice format (MISALIGNED). Figure 5 shows the effect of answer choice format alignment on the target task

performance by comparing ALIGNED and MIS-ALIGNED. The result shows that for all 3 datasets, ALIGNED significantly outperforms MISALIGNED. This result is non-trivial considering that the two prompt embeddings are trained on the same source training dataset and the same training configuration, with the only difference in the given answer choice format, implying that how the answer choices are given to solve a specific task is more important than the content of the training data for task location. ROSPR is mostly comparable to ALIGNED embedding, implying that retrieving a source prompt embedding by searching for similar input instances results in retrieving a source embedding with similar answer choice formats.

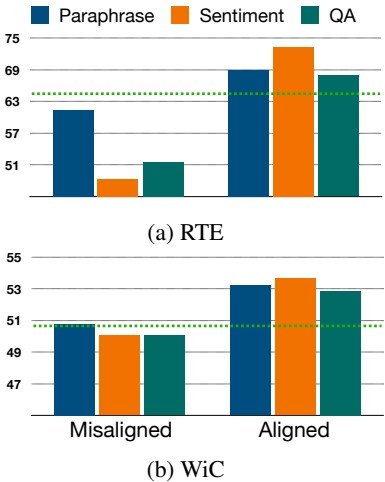

(a) RTE

(b) WiC

Figure 6: RTE (Top) and WiC (Bottom) evaluation result by retrieving MISALIGNED and ALIGNED answer choice format across various source tasks. We report the mean accuracy of the evaluation prompts and the performance of T0 is shown in green dotted line.

We additionally analyze the effect of answer choice formats on RTE and WiC datasets by retrieving prompt embeddings trained on various source tasks. Both target datasets have the answer choices of either yes/no. Similar to the previous experiments, we retrieve ALIGNED (yes/no format) and MISALIGNED prompt embeddings across three source tasks: paraphrase, sentiment classification, and multi-choice QA. As shown in Figure 6, for both target datasets, ALIGNED outperforms MISALIGNED across all three source tasks. This shows that aligning to the answer choice format of the target task is crucial regardless of the retrieved source task.

**Answer choice format is more important than task similarity.** From Figure 6, we can see that all three source tasks benefit from aligning to the target task answer choice format. One may think

that embeddings from source tasks requiring similar knowledge to the target task may be important. Counterintuitively, for both RTE and WiC target tasks, when the answer choice format is aligned, the task source embedding of sentiment classification, which is known to be irrelevant to RTE and WiC (Pruksachatkun et al., 2020), outperforms other embeddings sourced from datasets that are more relevant to the target datasets (paraphrase and multi-choice QA) (Appendix G). This implies that for retrieval of source embedding for task location, answer choice format is more important than containing similar knowledge required to solve the target task.

**Role of ROSPR is similar to in-context learning.** From the findings explained in previous paragraphs, we can conclude that although the source task types influence the target task performance, retrieving a similar *answer choice format* is more important for task location. Indeed, source tasks containing similar knowledge can help target tasks only if the answer choice formats are aligned to the target task. These findings support Min et al. (2022); Webson and Pavlick (2021) that a meta-trained LM "takes less effort" to understand the input: models exploit the simple aspects of prompts and demonstrations such as the format and distribution instead of complex semantics. Especially, for in-context learning, Xie et al. (2021); Min et al. (2022) show that the role of demonstrations lies in providing the shared concept and distribution hints of the target task. From this aspect, the role of ROSPR is similar to demonstrations. However, it is more efficient than including demonstrations because it avoids heavy computation at inference from long sequence lengths (Liu et al., 2022a; Choi et al., 2022) since ROSPR prepends a fixed length of prefix tokens regardless of the task. Also, ROSPR is free from the instability of in-context learning coming from different orderings of demonstrations (Lu et al., 2022; Zhao et al., 2021). Lastly, we conjecture that ROSPR also has the benefits of *soft* prompts (Li and Liang, 2021) such as having more expressiveness.

## 7 Ablation Studies

In this section, we analyze the effect of the number of (1) prompts, (2) source datasets, and (3) queries sampled for evaluation. We evaluate variations of our proposed methods on 4 datasets: RTE for NLI, COPA for sentence completion, Winogrande

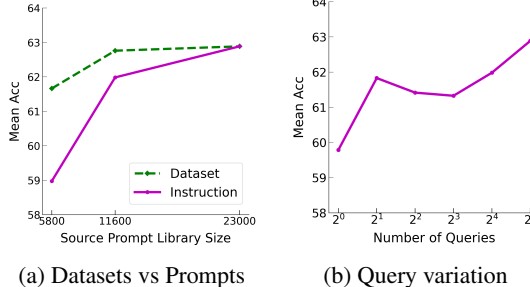

(a) Datasets vs Prompts          (b) Query variation

Figure 7: Result of various ablation settings of ROSPR. (a) compares the effect of scaling the number of datasets with scaling the number of prompts and (b) shows the effect of the number of sampled queries at inference. Additional ablation results are shown in Appendix D.

for coreference, and WiC for word sense disambiguation task. We report the average of the mean accuracy of all evaluation prompts for each dataset by running 3 different runs.

**Scaling number of prompts vs. number of datasets.** Recent works on meta-trained LMs show that the number of source datasets and prompts is an important factor for zero-shot task generalization (Sanh et al., 2021; Wei et al., 2021; Wang et al., 2022b; Chung et al., 2022). We also show ablations for ROSPR and measure how the zero-shot generalization performance changes when we vary the number of prompts and datasets available during the prompt tuning stage (shown in Figure 7a). First, we vary (1) the total number of source prompts by 60, 120, and 230 by increasing the number of prompts *per dataset* and (2) the number of datasets by 8, 16, and 30 by increasing the number of datasets *per task cluster*.[9] Note that in (1), the total number of datasets is fixed while in (2), we use all available prompts for each dataset while varying the number of datasets per task cluster.

In contrast to (1), (2) does not always lead to a linearly increasing performance boost; the performance saturates as more source datasets are included. By comparing the effect of scaling datasets and scaling prompts for similar Source Prompt Library sizes, we observe that the number of prompts has more impact on the accuracy of the target task (Figure 7a).

This ablation study also supports the analysis of the previous section; diverse answer choice formats of prompts, which are mostly influenced by the total number of source prompts, are more important than source task types which are influenced by the number of source datasets.[10] Therefore, if

the number of task clusters is sufficient to some extent, scaling the number of source prompts per dataset is more crucial than scaling the number of source datasets per task cluster.

**More sampled queries improve the performance.** We also analyze the effect of the number of query instances sampled at inference for retrieval. As seen in Figure 7b, increasing the number of queries results in higher mean accuracy. This is different from the analysis of Lin et al. (2022) that sampling more queries leads to better performance only to some point. Because we use the frequency of each prompt embedding candidate as the default metric for retrieval, utilizing more query instances would represent the evaluation data more accurately, resulting in a reduced number of wrong retrievals.

## 8 Conclusion

In this paper, we introduce ROSPR, a method that efficiently enhances zero-shot generalization capabilities of a meta-trained LM by retrieving prompt-specific source prompt embeddings (soft prompts) for a given target task. We accomplish this by first training the soft prompts for hard prompt of the source tasks. After training source prompt embeddings, we construct the *Source Prompt Library* by storing the mean representation of training instances as keys and the corresponding prompt embeddings as values. At inference, we search for training instances stored in the library similar to sample instances from the target task, retrieve the corresponding prompt embedding, select the most frequently retrieved embedding, and append it to each of the target task instances for prediction. Our results show that ROSPR efficiently enhances the zero-shot performance of the backbone model while introducing minimal additional parameters during inference. We additionally provide analysis of which factors attribute to the performance of ROSPR and find that *heuristic cues* such as the answer choice format are critical for generalization performance, implying that it may play a role similar to demonstrations in in-context learning.

## 9 Limitations

Although we show the effectiveness of ROSPR by applying it on T0-3B (Sanh et al., 2021), we did

---

[9]Task cluster is defined as a cluster of the same task types.

[10]Although the total number of prompts also increases as

the number of datasets increases, we find that answer choice formats are similar across the same task type, meaning that the diversity of answer choice formats is not increased by increasing the number of datasets per task clusters.

not evaluate our method on different model scales such as the T0-11B variant and other LM architectures such as decoder-only LMs due to limited computational resources. This leaves future works on applying ROSPR to even larger LMs and diverse LM architectures (Wang et al., 2022a). Moreover, it is hard to apply VAR to target tasks without answer choices such as free-form generation because variance among options cannot be obtained. However, ROSPR and ROSPR+INTER can still be utilized and we leave applying ROSPR on zero-shot task location of free-form generation as future work (Scialom et al., 2022).

## Acknowledgements

We thank Sohee Yang and Eunbi Choi for helpful feedback on the paper. This work was partly supported by Institute of Information & communications Technology Planning & Evaluation (IITP) grant funded by the Korea government (MSIT) (No.2022-0-00264, Comprehensive Video Understanding and Generation with Knowledge-based Deep Logic Neural Network, 80%; No.2019-0-00075, Artificial Intelligence Graduate School Program (KAIST), 20%).

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

## A Comparison with Hard Prompt Optimization

We have conducted additional experiments to compare our method with 3 different hard prompt optimization techniques including (1) RoHPr which utilizes the same search technique as RoSPr but retrieves 4 corresponding training instances for in-context learning instead of the corresponding soft prompt, and (2) APE (Zhou et al., 2023) which is an automatic prompt engineering method that utilizes the generation results of an LLM (Ouyang et al., 2022), known to show better performance than human instructions, and (3) ZPS (Liao et al., 2022) which selects an optimal hard prompt from a prompt pool using prompt ensemble and pseudo-labels. As shown in the result of Table 2, on average, RoSPr performs the best on average, showing the benefits of using soft prompts over hard prompts for task generalization. While RoSPr shows improvement on 10 out of 11 datasets, other hard prompt optimization methods do not show consistent improvements.

## B Full Result of BIG-bench Evaluation

We provide the task generalization performance result of 14 tasks from BIG-bench, shown in Table 3. Applying ROSPR largely improves the performance for 3 datasets (Hindu Knowledge, Novel Concepts, Misconceptions): (+17.72%, +3.12% +1.82%) compared to T0-3B and (+13.72%, +3.12%, +2.05%) compared to T0-11B. For mean accuracy of 14 tasks, T0+ROSPR outperforms T0-3B by 2.39% points, reducing the gap between 4 times larger T0-11B to 1.84% points. Moreover, applying INTER to T0+ROSPR enhances the performance of T0+ROSPR for most tasks, indicating that interpolation of multiple embeddings is effective for challenging tasks.

## C Fine-tuning Baseline Details

For fine-tuning baseline models (PT and AT-TEMPT), we follow the training configuration of source prompt tuning. We train each target prompt with a single epoch using a maximum of 5,000 training instances. For fine-tuning, we use a batch size of 32 and a learning rate of 1e-3. Also, we randomly select hard prompts (templates) of the training dataset during fine-tuning. For ATTEMPT, we randomly sample one source prompt per task cluster, resulting in interpolation between 8 soft prompts.

## D Additional Ablation Studies

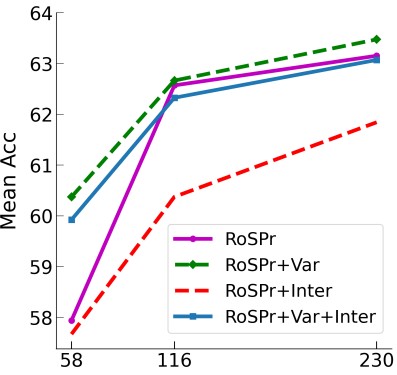

Figure 8: Variation of number of prompts by increasing the number of prompts per dataset.

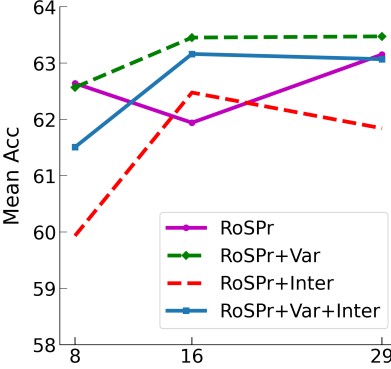

Figure 9: Variation of number of datasets by increasing the number of datasets per task cluster.

We provide detailed result of variation of the number of prompts (Figure 8) and the number of datasets (Figure 9). We additionally analyze the effect of (1) different sampling methods for constructing the Source Prompt Library, (2) the number of instances sampled for constructing the Source Prompt Library, (3) the number of top-N retrieval for embedding retrieval, and (4) the number of multiple source embeddings to interpolate. Same as Section 7, we report the mean accuracy of 4 evaluation datasets: RTE, COPA, Winogrande, and WiC with 3 runs with different random seeds for the sampling of evaluation queries.

### D.1 Sampling Methods for Source Prompt Library

We experiment three different methods to sample instances for constructing Source Prompt Library and analyze the effect of each method. By default, we choose RANDOM method, where we sample

| | RTE | CB | AN.R1 | AN.R2 | AN.R3 | COPA | Hellasw. | StoryC. | Winogr. | WSC | WiC | AVG |
|---|---|---|---|---|---|---|---|---|---|---|---|---|
| T0 | 64.55 | 45.36 | 33.81 | 33.11 | 33.33 | 75.88 | 26.60 | 84.03 | 50.97 | 65.10 | 50.69 | 51.22 |
| RoHPr | 47.10 | 45.00 | 33.53 | 33.30 | 33.09 | 73.38 | 26.65 | 86.93 | 50.51 | 64.42 | 50.00 | 49.45 |
| APE (Zhou et al., 2023) | 68.19 | 48.33 | 35.82 | 35.37 | 34.27 | 68.00 | 25.76 | 78.73 | 50.43 | 58.75 | 50.85 | 50.41 |
| ZPS (Liao et al., 2022) | 58.48 | 60.71 | 36.60 | 34.40 | 33.33 | 76.00 | 28.49 | 87.39 | 51.78 | 64.42 | 50.63 | 52.93 |
| RoSPr | 71.54 | 49.48 | 37.05 | 34.64 | 33.92 | 78.75 | 26.97 | 85.52 | 51.50 | 64.52 | 51.76 | **53.24** |

Table 2: Comparison result of RoSPr with different hard prompt optimization techniques.

| | T0-3B | ROSPR | INTER | T0-11B |
|---|---|---|---|---|
| Strategy. | **52.79** | 52.05 | 52.23 | 52.75 |
| Movie D. | 52.85 | 51.45 | 52.23 | **53.69** |
| Known Un. | 47.83 | 47.83 | 47.83 | **58.70** |
| Logic Grid | **41.10** | 37.00 | 37.40 | 38.30 |
| Hindu Kn. | 25.71 | 43.43 | **45.71** | 29.71 |
| Code D. | **46.67** | 45.00 | 40.00 | 43.33 |
| Concept | 45.52 | 67.61 | 67.61 | **69.29** |
| Language | 14.84 | 13.68 | 14.40 | **20.20** |
| Vitamin | 58.88 | 53.71 | 54.53 | **64.73** |
| Syllogism | **52.94** | 50.64 | 51.34 | 51.81 |
| Misconcept. | 50.23 | **52.05** | **52.05** | 50.00 |
| Logical | 46.64 | **54.86** | **54.86** | **54.86** |
| Winowhy | 44.29 | 44.33 | 44.29 | **52.11** |
| Novel Con. | 15.63 | 15.63 | **18.75** | 15.63 |
| AVG | 42.56 | 44.95 | 45.23 | **46.79** |

Table 3: Evaluation result of 14 tasks from BIG-bench (Srivastava et al., 2022).

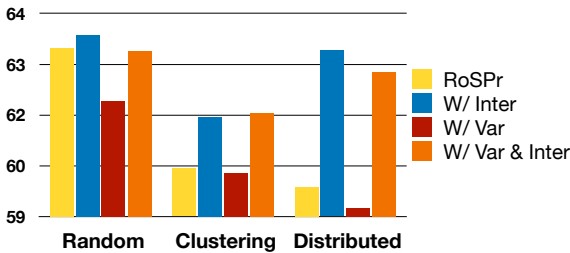

Figure 10: Different instance sampling methods for constructing Source Prompt Library.

100 instances by random for each prompt by assuming that each random 100 queries of instances can represent the whole prompt. Second, we experiment CLUSTERING method, where we sample top 100 instances which has the closest mean representation from its overall average for each prompt. If we say each instance has a distance of mean representation from its overall average as $d_i$ in increasing order, we sample $\{d_1, d_2, ..., d_{100}\}$. The last method we use is DISTRIBUTED method, where we sample 100 instances in a distributed way with respect to its distance of mean representation from its overall average. If we say each instance has a distance of mean representation from its over-

all average as $d_i$ in increasing order, we sample $\{d_1, d_{1+N/100}, d_{1+2*N/100}, ..., d_{1+99*N/100}\}$, assuming there are total $N$ training instances in a dataset.

As shown in Figure 10, RANDOM method outperforms CLUSTERING and DISTRIBUTED methods. Interestingly, CLUSTERING method significantly hurts the performance on all 4 proposed methods, suggesting that storing similar instances per prompt results in retrieval failures more often. Also for DISTRIBUTED method, most of the methods significantly underperform RANDOM, except INTER and ROSPR+VAR+INTER. From these results, we can conclude that random sampling represents the source dataset most effectively.

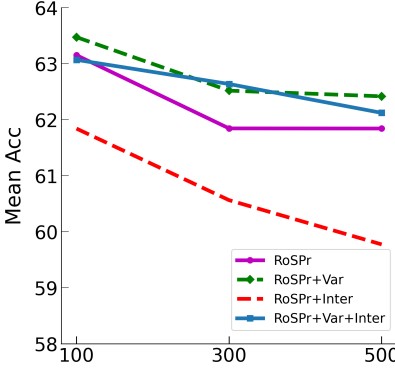

Figure 11: Variation of number of instances sampled for constructing Source Prompt Library. Default setting is $n = 100$.

### D.2 Number of Instances Sampled for Constructing Source Prompt Library

We analyze the effect of size of the Source Prompt Library by varying the number of instances $n$ to sample for each hard prompt by 100, 300, 500. Therefore, $n \times$ (number of total hard prompts) would be the size of the Source Prompt Library. As shown in Figure 11, increasing the number of sampled instances does not increase the performance; it hurts the performance for most cases. This suggests that only a few number of training instances are enough to represent the distribution of prompted

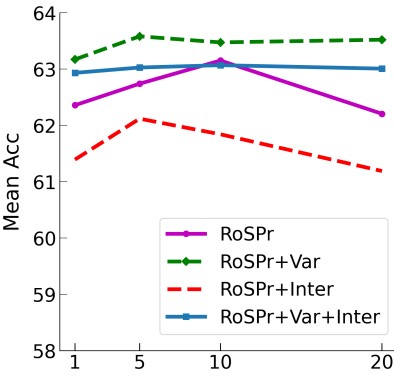

Figure 12: Variation of number of Top $N$ instances for embedding retrieval. Default setting is $N = 10$.

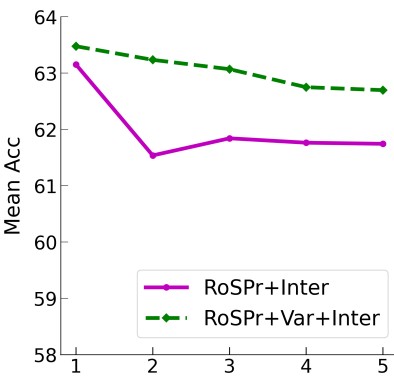

Figure 13: Variation of number of source embeddings for interpolation based methods. Default setting is $K' = 3$.

input (hard prompt + input instances) for each hard prompt and increasing the number sometimes hurt the performance by adding noise to the distribution. This also supports the importance of heuristic cues in Section 6 by showing that adding more training instances *per hard prompt* does not increase the performance. Instead, adding hard prompts with diverse *answer choice format* is more important.

### D.3 Number of Top N instances for Embedding Retrieval

We vary the number of top-$N$ instances that are retrieved given each query through MIPS search. As shown in Figure 12, varying the number of top-$N$ instances does not have much effect compared to increasing the number of sampled queries (Figure 7b). This implies that if the size of the evaluation set of the target task is large, sampling more queries is effective than searching for more similar instances per query. This is important for variance-based methods because the number of forward passes needed before evaluation is proportional to $Q * N$. Therefore, we can reduce the time latency by reducing the number of instances retrieved per query without hurting the performance much.

### D.4 Number of Source Embeddings for Interpolation

We analyze the effect of number of source embeddings for interpolation by varying top-$N'$ interpolation from 1 (no interpolation) to 5 shown in Figure 13. By comparing between single prompt embedding retrieval ($N' = 1$) and the interpolation of multiple embeddings ($N' > 1$), the mean accuracy drops by adding multiple source embeddings for retrieval because interpolation-based methods underperform on tasks such as COPA as shown

in Table 1. Mean accuracy would increase if we add other datasets for evaluation that benefit from interpolation such as WSC and CB.

By comparing among various $N'$ values, we find that for ROSPR+INTER, the accuracy substantially decreases for $K' = 2$, implying that the possibility of a wrong retrieval varies depending on the value of $N'$. In contrast, ROSPR+VAR+INTER is more robust to the value of $K'$, showing that variance-based ranking increases robustness to different numbers of source embeddings for interpolation as well.

### E Visualization of Results

We show the visualization of the evaluation result on 11 datasets in Figure 14. Methods based on ROSPR not only show higher accuracy, but lower variance for many datasets.

### F Experimental Configurations

As mentioned in the previous sections, we use T0-3B as our backbone meta-trained LM. For prompt tuning, we fix the prefix length as 100 and the embeddings are initialized from 5,000 most common vocabulary following Lester et al. (2021). We train each source embedding for a single epoch with a learning rate of 0.1 and a batch size of 32. We use the Adam optimizer with weight decay of 1e-5. For retrieval, we randomly sample $Q = 32$ query instances and retrieve top $K = 10$ examples for each query. We train a T0-small variant ($\sim 35$M params) as our dense retriever by multi-task prompted training on T5+LM model (Lester et al., 2021), replicating the original training setting of T0 by training T5+LM for 8 epochs using the same training instances of Sanh et al. (2021) with

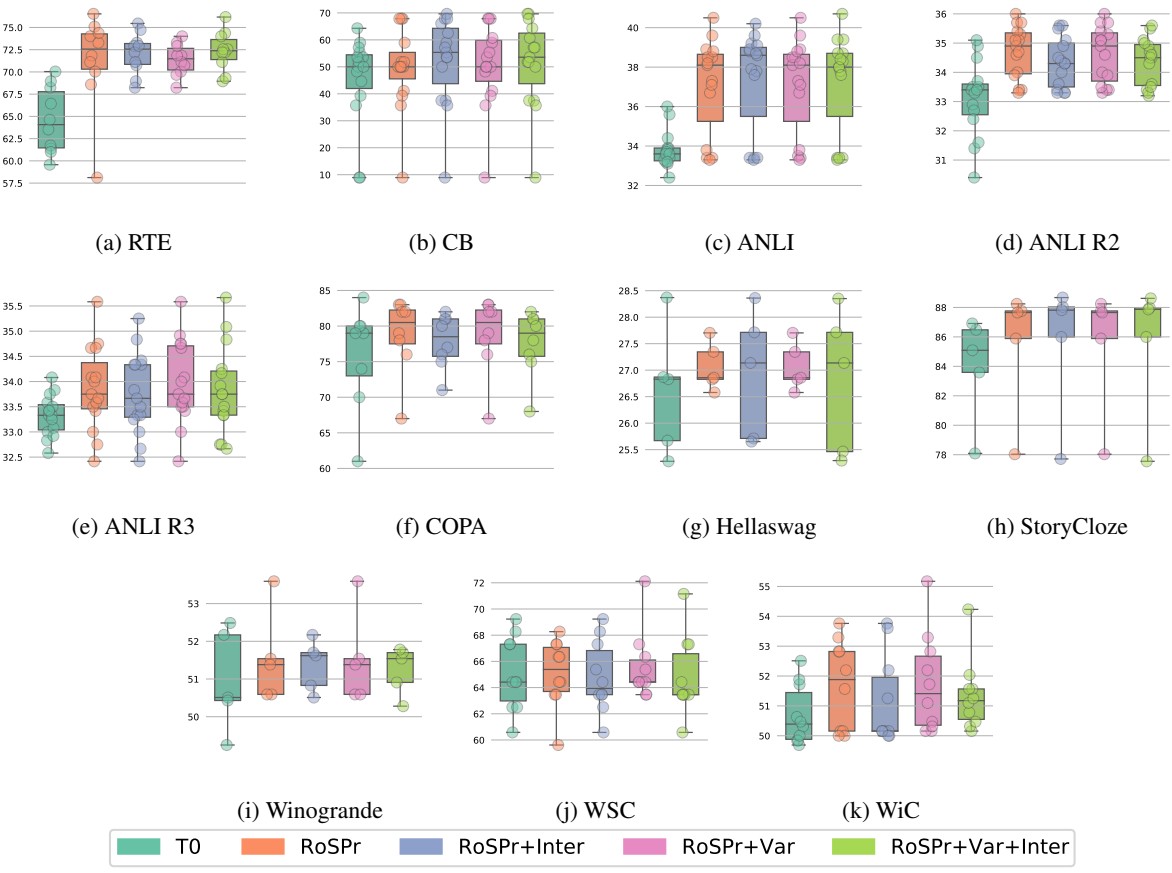

(a) RTE     (b) CB     (c) ANLI     (d) ANLI R2

(e) ANLI R3     (f) COPA     (g) Hellaswag     (h) StoryCloze

(i) Winogrande     (j) WSC     (k) WiC

T0     RoSPr     RoSPr+Inter     RoSPr+Var     RoSPr+Var+Inter

Figure 14: Visualization of evaluation results of 11 datasets.

a learning rate of 1e-3, input sequence length 512, output sequence 128, and batch size of 1024. We select model checkpoint by early stopping based on validation accuracy. We use a meta-trained LM instead of a naive pretrained model (e.g. Sentence-BERT) because meta-trained LM is shown to be more effective for retrieval (Lin et al., 2022). For the interpolation experiment, we set $K' = 3$ for top-$K'$ prompt embedding candidates. For training source prompt embeddings, we used 8 V100 GPUs.

## G    Examples of Applying Prompts, Answer Choice Format and Source Task Types

Figure 15 shows an example of applying prompt through Promptsource (Bach et al., 2022) as mentioned in Section 3.1.

We assert that *answer choice format* is more important than task similarity in Section 6. We further provide details of the input instances of the mentioned tasks: paraphrase, NLI, word sense disambiguation, and sentiment classification in Figure 16. As supported in Pruksachatkun et al. (2020),

intuitively, paraphrase task is more similar to the task of word sense disambiguation task or NLI, implying their task similarity, while the task of sentiment classification is very different. However, our counterintuitive result shows the *soft* prompt to show the best performance in Figure 6 Section 6, bolstering the claim that similar source task types are not a major factor for evaluation performance.

## H    Full List of Source Training and Evaluation Datasets

All of our training and evaluation datasets are a subset of datasets used in Sanh et al. (2021). We use Huggingface version of each dataset (Lhoest et al., 2021).

### H.1    Training Datasets

Following Sanh et al. (2021), we use 8 task clusters for training of source prompt embedding: sentiment classification, paraphrase, topic classification, summarization, struc-to-text, multiple-choice QA, extractive QA, and closed book QA. We use imdb (Maas et al., 2011), amazon_polarity

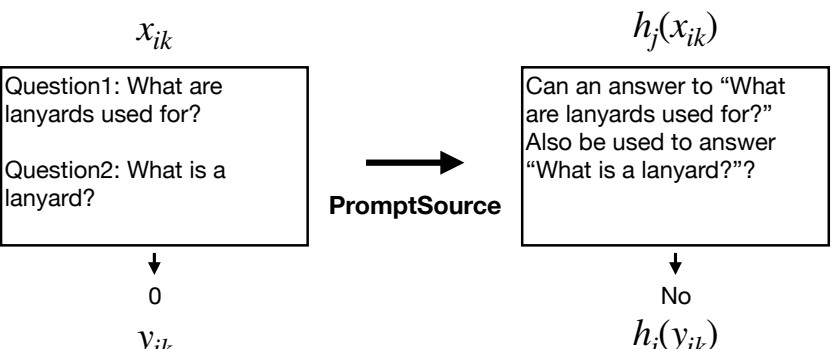

$$x_{ik}$$

Question1: What are lanyards used for?

Question2: What is a lanyard?

$$\downarrow$$

0

$$y_{ik}$$

$$h_j(x_{ik})$$

Can an answer to "What are lanyards used for?" Also be used to answer "What is a lanyard?"?

$$\downarrow$$

No

$$h_j(y_{ik})$$

**PromptSource**

Figure 15: Example of applying prompt to a given instance through Promptsource (Bach et al., 2022).

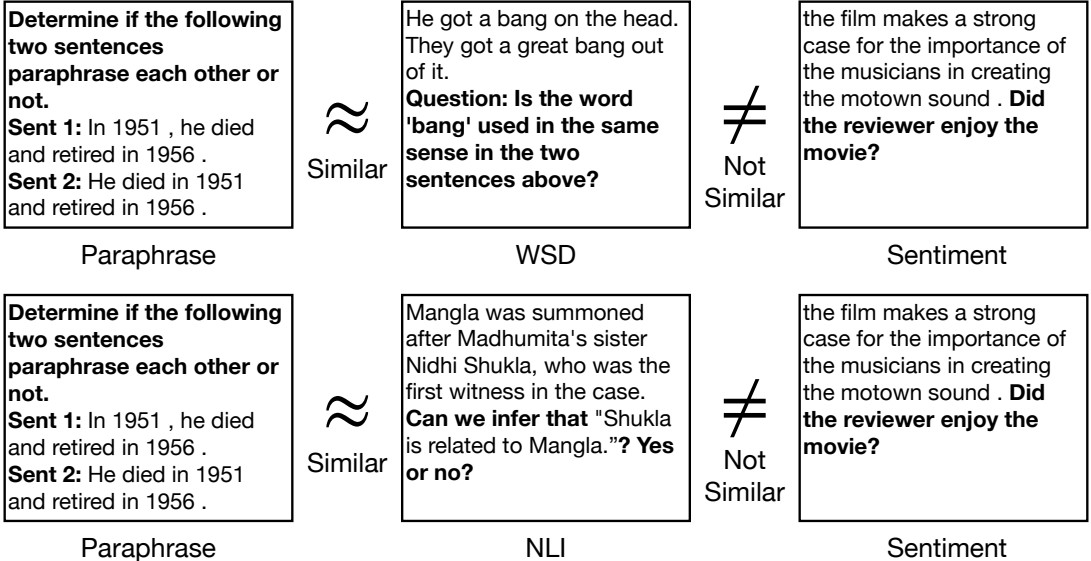

**Determine if the following two sentences paraphrase each other or not.**
**Sent 1:** In 1951 , he died and retired in 1956 .
**Sent 2:** He died in 1951 and retired in 1956 .

Paraphrase

≈ Similar

He got a bang on the head. They got a great bang out of it.
**Question: Is the word 'bang' used in the same sense in the two sentences above?**

WSD

≠ Not Similar

the film makes a strong case for the importance of the musicians in creating the motown sound . **Did the reviewer enjoy the movie?**

Sentiment

**Determine if the following two sentences paraphrase each other or not.**
**Sent 1:** In 1951 , he died and retired in 1956 .
**Sent 2:** He died in 1951 and retired in 1956 .

Paraphrase

≈ Similar

Mangla was summoned after Madhumita's sister Nidhi Shukla, who was the first witness in the case. **Can we infer that** "Shukla is related to Mangla."**? Yes or no?**

NLI

≠ Not Similar

the film makes a strong case for the importance of the musicians in creating the motown sound . **Did the reviewer enjoy the movie?**

Sentiment

Figure 16: Examples of instances of different source tasks.

(McAuley and Leskovec, 2013), rotten_tomatoes (Pang and Lee, 2005), yelp_review_full (Zhang et al., 2015b) for sentiment, glue/qqp (Wang et al., 2018), paws/labeled_final (Zhang et al., 2019) for paraphrase, ag_news (Zhang et al., 2015a), dbpedia_14 (Lehmann et al., 2015) for topic classification, gigaword (Graff et al., 2003), multi_news citefabbri-etal-2019-multi, samsum (Gliwa et al., 2019), xsum (Narayan et al., 2018) for summarization, common_gen (Lin et al., 2020), wiki_bio (Lebret et al., 2016) for struct-to-text, cos_e/v1.11 (Rajani et al., 2019), quail (Rogers et al., 2020), social_i_qa (Sap et al., 2019), wiqa (Tandon et al., 2019), cosmos_qa (Huang et al., 2019), sciq (Welbl et al., 2017), wiki_hop/original (Welbl et al., 2018) for multi-choice QA, adversarial_qa/dbidaf, adversarial_qa/dbert, adversarial_qa/droberta, quoref (Bartolo et al., 2020), ropes (Lin et al., 2019), duorc/SelfRC, duorc/Paraphrase IdentificationRC (Saha et al., 2018) for extractive QA, and kilt_tasks/hotpotqa (Petroni et al., 2021), wiki_qa (Yang et al., 2015) for closed book QA.

We exclude 6 datasets (MRPC, TREC, DREAM, QuaRTz, QASC, QuaRel) that have small training sets because it leads to task imbalance, which is critical for training our small dense retriever ($\sim$ 35M params). We also exclude CNN Daily Mail, App Reviews, and WikiQA dataset due to dataset download issues, absence of any test or validation data, and unbalanced label distribution, respectively.

## H.2 Evaluation Datasets

Following Sanh et al. (2021), we include 11 evaluation datasets as follows: RTE (Dagan et al., 2005), CB (De Marneffe et al., 2019), ANLI (Nie et al., 2020) for natural language inference task, COPA (Roemmele et al., 2011), Hellaswag (Zellers et al., 2019), Storycloze (Mostafazadeh et al., 2016) for sentence completion task, Winogrande (Sakaguchi et al., 2021), WSC (Levesque et al., 2012) for coreference resolution task, and WiC (Pilehvar and Camacho-Collados, 2019) for word sense disambiguation task.

For BIG-bench tasks, we evaluate on 14 tasks, following Sanh et al. (2021) : Known Unknown, Logic Grid, StrategyQA, Hindu Knowledge, Movie Dialog, Code Description, Conceptual, Language ID, Vitamin C, Syllogisms, Misconceptions, Logical Deduction, Winowhy and Novel Concepts.

## I Full List of Retrieved Prompt Embeddings

We provide a full list of retrieved prompt embeddings of RoSPR and ORACLE for all prompts of 11 evaluation datasets. We report retrieval results of a single random seed (Table 4 $\sim$ Table 14).

| Prompt Name | T0 | RoSPr | Retrieved Embedding | Oracle | Retrieved Embedding |
|---|---|---|---|---|---|
| GPT-3 style | 61.37 | 74.01 | paws/labeled_final/PAWS-ANLI GPT3 | 74.01 | paws/labeled_final/context-question |
| MNLI crowdsource | 63.53 | 70.04 | paws/labeled_final/context-question | 72.20 | paws/labeled_final/context-question |
| based on the previous passage | 68.23 | 76.53 | paws/labeled_final/context-question | 76.53 | paws/labeled_final/PAWS-ANLI GPT3 |
| can we infer | 59.57 | 73.29 | glue/qqp/meaning | 73.29 | paws/labeled_final/PAWS-ANLI GPT3 |
| does it follow that | 61.73 | 71.84 | paws/labeled_final/context-question | 71.84 | paws/labeled_final/context-question-no-label |
| does this imply | 64.62 | 68.59 | paws/labeled_final/context-question | 71.48 | paws/labeled_final/context-question |
| guaranteed true | 68.95 | 75.09 | paws/labeled_final/context-question | 75.81 | paws/labeled_final/PAWS-ANLI GPT3 |
| should assume | 66.43 | 71.12 | glue/qqp/meaning | 76.53 | paws/labeled_final/PAWS-ANLI GPT3 |
| justified in saying | 61.01 | 58.12 | paws/labeled_final/paraphrase-task | 71.12 | paws/labeled_final/context-question |
| must be true | 70.04 | 74.37 | paws/labeled_final/context-question | 75.09 | paws/labeled_final/PAWS-ANLI GPT3-no-label |
| **Avg.** | **64.55** | **71.30** | | **73.79** | |

Table 4: List of retrieved source prompts of RoSPr and Oracle for each evaluation prompts of RTE.

| Prompt Name | T0 | RoSPr | Retrieved Embedding | Oracle | Retrieved Embedding |
|---|---|---|---|---|---|
| can we infer | 55.36 | 51.79 | social_i_qa/Show choices and generate index | 67.86 | samsum/Write a dialogue that match this summary |
| based on the previous passage | 44.64 | 58.93 | social_i_qa/Show choices and generate index | 69.64 | samsum/Write a dialogue that match this summary |
| claim true/false/inconclusive | 50.00 | 67.86 | social_i_qa/Show choices and generate index | 69.64 | samsum/Summarize: |
| does it follow that | 64.29 | 50.00 | social_i_qa/Show choices and generate index | 67.86 | samsum/Summarize: |
| justified in saying | 53.57 | 50.00 | social_i_qa/Show choices and generate index | 62.50 | samsum/Write a dialogue that match this summary |
| always/sometimes/never | 39.29 | 41.07 | social_i_qa/Show choices and generate index | 41.07 | social_i_qa/Show choices and generate index |
| GPT-3 style | 51.79 | 67.86 | social_i_qa/Show choices and generate index | 69.64 | social_i_qa/Show choices and generate answer |
| consider always/sometimes/never | 35.71 | 35.71 | social_i_qa/Show choices and generate index | 39.29 | social_i_qa/Generate answer |
| guaranteed true | 48.21 | 50.00 | social_i_qa/Show choices and generate index | 64.29 | social_i_qa/Generate answer |
| must be true | 53.57 | 50.00 | social_i_qa/Show choices and generate index | 64.29 | social_i_qa/Generate answer |
| guaranteed/possible/impossible | 8.93 | 8.93 | social_i_qa/Show choices and generate index | 8.93 | -(all same) |
| does this imply | 58.93 | 51.79 | social_i_qa/Show choices and generate index | 66.07 | glue/qqp/duplicate |
| MNLI crowdsource | 8.93 | 39.29 | social_i_qa/Show choices and generate index | 42.86 | cos_e/v1.11/question_option_description_text |
| should assume | 57.14 | 50.00 | social_i_qa/Show choices and generate index | 66.07 | cos_e/v1.11/aligned_with_common_sense |
| take the following as truth | 50.00 | 67.86 | social_i_qa/Show choices and generate index | 71.43 | cos_e/v1.11/description_question_option_id |
| **Avg.** | **45.36** | **49.40** | | **58.10** | |

Table 5: List of retrieved source prompts of RoSPr and Oracle for each evaluation prompts of CB.

| Prompt Name | T0 | RoSPr | Retrieved Embedding | Oracle | Retrieved Embedding |
|---|---|---|---|---|---|
| can we infer | 33.90 | 38.90 | paws/labeled_final/context-question | 39.40 | paws/labeled_final/PAWS-ANLI GPT3 |
| based on the previous passage | 33.90 | 38.50 | paws/labeled_final/context-question | 38.60 | paws/labeled_final/PAWS-ANLI GPT3 |
| claim true/false/inconclusive | 35.60 | 36.70 | paws/labeled_final/PAWS-ANLI GPT3 | 39.10 | paws/labeled_final/context-question |
| does it follow that | 36.00 | 40.50 | paws/labeled_final/context-question | 40.50 | paws/labeled_final/PAWS-ANLI GPT3 |
| justified in saying | 33.10 | 38.10 | paws/labeled_final/context-question | 38.80 | paws/labeled_final/context-question-no-label |
| always/sometimes/never | 33.40 | 33.40 | paws/labeled_final/paraphrase-task | 33.40 | -(all 33.4) |
| GPT-3 style | 33.80 | 37.30 | paws/labeled_final/PAWS-ANLI GPT3 | 38.50 | paws/labeled_final/PAWS-ANLI GPT3-no-label |
| consider always/sometimes/never | 33.20 | 33.40 | paws/labeled_final/PAWS-ANLI GPT3 | 33.50 | -(all 33.4) |
| guaranteed true | 33.70 | 38.50 | paws/labeled_final/context-question | 38.70 | paws/labeled_final/PAWS-ANLI GPT3 |
| must be true | 34.40 | 39.60 | paws/labeled_final/context-question | 39.70 | paws/labeled_final/context-question |
| guaranteed/possible/impossible | 33.30 | 33.30 | paws/labeled_final/PAWS-ANLI GPT3 | 33.30 | imdb/Text Expressed Sentiment |
| does this imply | 33.60 | 38.20 | paws/labeled_final/context-question | 38.20 | paws/labeled_final/context-question-no-label |
| MNLI crowdsource | 33.60 | 33.80 | paws/labeled_final/context-question | 35.40 | dbpedia_14/given_list_what_category_does_the_paragraph_belong_to |
| should assume | 33.20 | 38.80 | paws/labeled_final/context-question | 39.00 | paws/labeled_final/PAWS-ANLI GPT3-no-label |
| take the following as truth | 32.40 | 37.10 | paws/labeled_final/PAWS-ANLI GPT3 | 38.70 | paws/labeled_final/context-question-no-label |
| **Avg.** | **33.81** | **37.07** | | **37.65** | |

Table 6: List of retrieved source prompts of RoSPr and Oracle for each evaluation prompts of ANLI R1.

| Prompt Name | T0 | RoSPr | Retrieved Embedding | Oracle | Retrieved Embedding |
|---|---|---|---|---|---|
| can we infer | 30.40 | 35.30 | paws/labeled_final/context-question | 35.30 | paws/labeled_final/PAWS-ANLI GPT3 |
| based on the previous passage | 31.40 | 35.40 | paws/labeled_final/context-question | 35.40 | paws/labeled_final/PAWS-ANLI GPT3 |
| claim true/false/inconclusive | 34.90 | 35.10 | paws/labeled_final/PAWS-ANLI GPT3 | 35.90 | paws/labeled_final/context-question |
| does it follow that | 34.50 | 36.00 | paws/labeled_final/context-question | 36.00 | paws/labeled_final/PAWS-ANLI GPT3 |
| justified in saying | 33.50 | 35.70 | paws/labeled_final/context-question | 35.70 | rotten_tomatoes/Reviewer Enjoyment Yes No |
| always/sometimes/never | 33.40 | 33.40 | paws/labeled_final/paraphrase-task | 33.50 | adversarial_qa/droberta/based_on_ |
| GPT-3 style | 33.50 | 34.90 | paws/labeled_final/PAWS-ANLI GPT3 | 35.00 | paws/labeled_final/context-question-no-label |
| consider always/sometimes/never | 33.70 | 33.40 | dbpedia_14/given_a_choice_of_categories | 34.50 | paws/labeled_final/PAWS-ANLI GPT3-no-label |
| guaranteed true | 32.90 | 34.00 | paws/labeled_final/context-question | 34.30 | paws/labeled_final/PAWS-ANLI GPT3 |
| must be true | 35.10 | 34.60 | paws/labeled_final/context-question | 35.10 | paws/labeled_final/PAWS-ANLI GPT3 |
| guaranteed/possible/impossible | 33.30 | 33.30 | paws/labeled_final/context-question-no-label | 33.30 | imdb/Writer Expressed Sentiment |
| does this imply | 32.70 | 33.90 | paws/labeled_final/context-question | 34.10 | paws/labeled_final/context-question |
| MNLI crowdsource | 33.40 | 34.70 | dbpedia_14/given_a_choice_of_categories | 34.90 | dbpedia_14/given_a_choice_of_categories |
| should assume | 32.40 | 35.10 | paws/labeled_final/context-question | 35.10 | paws/labeled_final/PAWS-ANLI GPT3 |
| take the following as truth | 31.60 | 35.70 | paws/labeled_final/paraphrase-task | 35.70 | paws/labeled_final/PAWS-ANLI GPT3 |
| **Avg.** | **33.11** | **34.70** | | **34.92** | |

Table 7: List of retrieved source prompts of RoSPr and Oracle for each evaluation prompts of ANLI R2.

| Prompt Name | T0 | RoSPr | Retrieved Embedding | Oracle | Retrieved Embedding |
|---|---|---|---|---|---|
| can we infer | 33.00 | 34.75 | glue/qqp/meaning | 34.75 | paws/labeled_final/context-question-no-label |
| based on the previous passage | 33.33 | 34.08 | paws/labeled_final/context-question | 35.33 | paws/labeled_final/context-question-no-label |
| claim true/false/inconclusive | 32.83 | 35.58 | paws/labeled_final/paraphrase-task | 35.92 | cos_e/v1.11/aligned_with_common_sense |
| does it follow that | 34.08 | 34.67 | paws/labeled_final/context-question | 35.33 | amazon_polarity/User_recommend_this_product |
| justified in saying | 33.58 | 33.00 | paws/labeled_final/paraphrase-task | 35.42 | amazon_polarity/User_recommend_this_product |
| always/sometimes/never | 33.42 | 33.42 | paws/labeled_final/paraphrase-task | 33.50 | paws/labeled_final/paraphrase-task |
| GPT-3 style | 33.33 | 34.00 | paws/labeled_final/PAWS-ANLI GPT3 | 34.92 | paws/labeled_final/PAWS-ANLI GPT3 |
| consider always/sometimes/never | 33.08 | 32.42 | ropes/plain_no_background | 33.67 | cos_e/v1.11/question_description_option_text |
| guaranteed true | 32.58 | 34.08 | paws/labeled_final/context-question | 34.83 | paws/labeled_final/context-question |
| must be true | 33.83 | 33.75 | paws/labeled_final/paraphrase-task | 35.42 | rotten_tomatoes/Reviewer Enjoyment Yes No |
| guaranteed/possible/impossible | 33.50 | 33.50 | paws/labeled_final/paraphrase-task | 33.58 | social_i_qa/Show choices and generate answer |
| does this imply | 32.92 | 33.58 | glue/qqp/meaning | 34.50 | paws/labeled_final/context-question |
| MNLI crowdsource | 33.75 | 33.67 | rotten_tomatoes/Text Expressed Sentiment | 34.00 | dbpedia_14/given_a_choice_of_categories |
| should assume | 33.25 | 34.67 | paws/labeled_final/context-question | 34.92 | paws/labeled_final/context-question-no-label |
| take the following as truth | 33.42 | 32.75 | social_i_qa/Show choices and generate index | 36.67 | paws/labeled_final/context-question-no-label |
| **Avg.** | **33.33** | **33.86** | | **34.91** | |

Table 8: List of retrieved source prompts of RoSPr and Oracle for each evaluation prompts of ANLI R3.

| Prompt Name | T0 | RoSPr | Retrieved Embedding | Oracle | Retrieved Embedding |
|---|---|---|---|---|---|
| exercise | 80.00 | 79.00 | cos_e/v1.11/question_option_description_text | 80.00 | cos_e/v1.11/question_option_description_text |
| plausible_alternatives | 84.00 | 83.00 | cos_e/v1.11/question_option_description_text | 83.00 | cos_e/v1.11/question_option_description_text |
| "C1 or C2? premise, so/because…" | 61.00 | 67.00 | social_i_qa/Check if a random answer is valid or not | 75.00 | cos_e/v1.11/description_question_option_text |
| best_option | 70.00 | 76.00 | social_i_qa/Show choices and generate answer | 79.00 | cos_e/v1.11/description_question_option_text |
| more likely | 79.00 | 83.00 | cos_e/v1.11/question_option_description_text | 85.00 | cos_e/v1.11/description_question_option_text |
| cause_effect | 74.00 | 78.00 | cos_e/v1.11/question_option_description_text | 83.00 | common_gen/random task template prompt |
| choose | 80.00 | 82.00 | cos_e/v1.11/question_option_description_text | 82.00 | cosmos_qa/no_prompt_text |
| i_am_hesitating | 79.00 | 82.00 | cos_e/v1.11/question_option_description_text | 82.00 | cos_e/v1.11/question_option_description_text |
| **Avg.** | **75.88** | **78.75** | | **81.13** | |

Table 9: List of retrieved source prompts of RoSPr and Oracle for each evaluation prompts of COPA.

| Prompt Name | T0 | RoSPr | Retrieved Embedding | Oracle | Retrieved Embedding |
|---|---|---|---|---|---|
| Predict ending with hint | 26.83 | 27.70 | social_i_qa/Show choices and generate index | 29.11 | cos_e/v1.11/question_option_description_text |
| Randomized prompts template | 26.87 | 26.83 | social_i_qa/Show choices and generate index | 27.84 | wiqa/what_is_the_missing_first_step |
| complete_first_then | 28.37 | 27.35 | ropes/prompt_bottom_no_hint | 28.35 | cos_e/v1.11/question_option_description_text |
| if_begins_how_continues | 25.28 | 26.58 | social_i_qa/Show choices and generate index | 26.58 | cos_e/v1.11/question_option_description_text |
| how_ends | 25.67 | 26.86 | social_i_qa/Show choices and generate index | 26.86 | cos_e/v1.11/question_option_description_text |
| **Avg.** | **26.60** | **27.06** | | **27.75** | |

Table 10: List of retrieved source prompts of RoSPr and Oracle for each evaluation prompts of Hellaswag.

| Prompt Name | T0 | RoSPr | Retrieved Embedding | Oracle | Retrieved Embedding |
|---|---|---|---|---|---|
| Answer Given options | 86.48 | 87.76 | social_i_qa/Show choices and generate answer | 87.76 | social_i_qa/Show choices and generate answer |
| Choose Story Ending | 86.91 | 88.24 | social_i_qa/Show choices and generate answer | 88.24 | cos_e/v1.11/question_option_description_text |
| Movie What Happens Next | 78.09 | 78.03 | social_i_qa/Show choices and generate answer | 87.33 | cosmos_qa/no_prompt_text |
| Story Continuation and Options | 83.59 | 85.89 | social_i_qa/Show choices and generate answer | 86.53 | social_i_qa/Show choices and generate answer |
| Novel Correct Ending | 85.09 | 87.65 | social_i_qa/Show choices and generate answer | 87.97 | social_i_qa/Show choices and generate index |
| **Avg.** | **84.03** | **85.52** | | **87.57** | |

Table 11: List of retrieved source prompts of RoSPr and Oracle for each evaluation prompts of StoryCloze.

| Prompt Name | T0 | RoSPr | Retrieved Embedding | Oracle | Retrieved Embedding |
|---|---|---|---|---|---|
| does underscore refer to | 49.25 | 51.54 | cos_e/v1.11/question_option_description_text | 51.78 | paws/labeled_final/PAWS-ANLI GPT3 |
| stand for | 50.51 | 50.59 | cos_e/v1.11/question_option_description_text | 52.09 | ropes/plain_no_background |
| underscore refer to | 50.43 | 50.59 | cos_e/v1.11/question_option_description_text | 52.33 | ropes/prompt_bottom_no_hint |
| fill in the blank | 52.17 | 51.38 | cos_e/v1.11/question_description_option_text | 52.01 | cos_e/v1.11/question_description_option_text |
| Replace | 52.49 | 53.59 | cos_e/v1.11/question_option_description_text | 53.59 | paws/labeled_final/paraphrase-task |
| **Avg.** | **50.97** | **51.54** | | **52.36** | |

Table 12: List of retrieved source prompts of RoSPr and Oracle for each evaluation prompts of Winogrande.

| Prompt Name | T0 | RoSPr | Retrieved Embedding | Oracle | Retrieved Embedding |
|---|---|---|---|---|---|
| does the pronoun refer to | 68.27 | 66.34 | social_i_qa/Check if a random answer is valid or not | 66.35 | paws/labeled_final/PAWS-ANLI GPT3 |
| by p they mean | 62.50 | 66.35 | social_i_qa/Check if a random answer is valid or not | 69.23 | glue/qqp/duplicate or not |
| in other words | 67.31 | 67.31 | social_i_qa/Show choices and generate answer | 72.12 | samsum/Write a dialogue that match this summary |
| I think they mean | 69.23 | 68.27 | social_i_qa/Show choices and generate answer | 78.08 | social_i_qa/Generate answer |
| replaced with | 64.42 | 59.62 | social_i_qa/Check if a random answer is valid or not | 66.35 | social_i_qa/Show choices and generate index |
| p is/are r | 62.50 | 64.42 | social_i_qa/Show choices and generate index | 65.38 | social_i_qa/Show choices and generate index |
| the pronoun refers to | 64.42 | 64.42 | social_i_qa/Show choices and generate index | 67.31 | social_i_qa/Check if a random answer is valid or not |
| Who or what is/are | 64.42 | 63.46 | social_i_qa/Show choices and generate index | 65.38 | samsum/To sum up this dialog |
| does p stand for | 67.31 | 63.46 | social_i_qa/I was wondering | 69.23 | samsum/Write a dialogue that match this summary |
| GPT-3 Style | 60.58 | 67.31 | social_i_qa/Show choices and generate index | 67.31 | rotten_tomatoes/Reviewer Expressed Sentiment |
| **Avg.** | **65.10** | **65.10** | | **68.17** | |

Table 13: List of retrieved source prompts of RoSPr and Oracle for each evaluation prompts of WSC.

| Prompt Name | T0 | RoSPr | Retrieved Embedding | Oracle | Retrieved Embedding |
|---|---|---|---|---|---|
| question-context-meaning-with-label | 50.31 | 53.76 | glue/qqp/meaning | 53.76 | paws/labeled_final/context-question-no-label |
| question-context-meaning | 50.63 | 50.16 | ropes/prompt_bottom_no_hint | 57.05 | paws/labeled_final/PAWS-ANLI GPT3 |
| grammar_homework | 49.84 | 50.16 | ropes/prompt_bottom_no_hint | 57.99 | rotten_tomatoes/Reviewer Enjoyment Yes No |
| affirmation_true_or_false | 49.69 | 51.57 | paws/labeled_final/Rewrite-no-label | 53.92 | paws/labeled_final/Meaning-no-label |
| GPT-3-prompt | 51.72 | 50.00 | social_i_qa/Show choices and generate index | 55.64 | paws/labeled_final/PAWS-ANLI GPT3 |
| same_sense | 49.84 | 52.82 | paws/labeled_final/Rewrite | 55.17 | paws/labeled_final/task_description-no-label |
| question-context | 51.88 | 53.29 | social_i_qa/Check if a random answer is valid or not | 57.05 | amazon_polarity/Is_this_product_review_positive |
| GPT-3-prompt-with-label | 50.47 | 52.19 | glue/qqp/meaning | 52.66 | glue/qqp/meaning |
| polysemous | 50.00 | 52.82 | paws/labeled_final/Rewrite | 53.29 | paws/labeled_final/Rewrite-no-label |
| similar-sense | 52.51 | 50.00 | social_i_qa/Show choices and generate index | 56.11 | paws/labeled_final/task_description-no-label |
| **Avg.** | **50.69** | **51.68** | | **55.26** | |

Table 14: List of retrieved source prompts of RoSPr and Oracle for each evaluation prompts of WiC.