# OpenReview forum: "Efficiently Enhancing Zero-Shot Performance of Instruction Following Model via Retrieval of Soft Prompt"
_EMNLP/2023/Conference — EMNLP 2023 Findings_

### Official Review · Reviewer_dsfs · 2023-08-05

**Soundness:** 2

**Excitement:**

2: Mediocre: This paper makes marginal contributions (vs non-contemporaneous work), so I would rather not see it in the conference.

**Paper Topic And Main Contributions:**

This paper aims to retrieve soft prompts to assist hard prompts in zero-shot task generalization. The authors train soft prompts for each hard prompt through prompt tuning, resulting in the creation of a source prompt library. During inference, the soft prompt embedding of the training instance that exhibits the highest similarity to the query instance is retrieved. The experimental evaluation encompasses a range of tasks, including natural language inference, sentence completion, coreference resolution, and word sense disambiguation. The paper also assesses the performance on BIG-Bench tasks.

**Reasons To Accept:**

The paper is well-crafted, providing clear and easily understandable logic.

The implementation details are presented clearly.

**Reasons To Reject:**

1. The idea of retrieval soft prompts is not novel.
2. The benefits of employing retrieval soft prompt embeddings to enhance zero-shot generalization have not been explicitly clarified. A comparison should be made with other methods that utilize retrieval hard prompts/examples for in-context learning.
3. The process of collecting a source prompt library necessitates fine-tuning the same backbone model to generate soft prompt embeddings. However, this poses a challenge for large-sized language models as fine-tuning becomes unfeasible, thus limiting the applicability of the method.


**Reproducibility:**

4: Could mostly reproduce the results, but there may be some variation because of sample variance or minor variations in their interpretation of the protocol or method.

**Reviewer Confidence:**

3: Pretty sure, but there's a chance I missed something. Although I have a good feel for this area in general, I did not carefully check the paper's details, e.g., the math, experimental design, or novelty.

---

> ### Author Rebuttal · Authors · 2023-08-29
>
> Thank you for making the effort to review our paper. We appreciate your recognition of the strengths of the paper.
> We think that all concerns or questions could be addressed. Our response is provided below.
>
> ---
> **The novelty of the technique**
>
> Although the idea of retrieval of soft prompts has been explored in previous works [[SPoT (Vu et al., ACL 2022)](https://arxiv.org/abs/2110.07904), [ATTEMPT (Asai et al., EMNLP 2022)](https://arxiv.org/abs/2205.11961)], our work has technical novelty in the following aspects:
> - Our work is the first to observe the effect of retrieval of soft prompt on the **zero-shot** task generalization setting where the training data of the target task does not exist by applying the retrieval process to an instruction-tuned model and showing better performance than hard prompt optimization techniques (see the additional table below) and previous approaches that require target task fine-tuning (PT and ATTEMPT performance on Table 1 of the paper).
> - Other than the retrieval of soft prompt, we additionally introduce Var (variance-based ranking) and Inter+Var (interpolation between soft prompts searched through variance-based ranking) which improve the mean accuracy and reduce prompt sensitivity (STD) of RoSPr shown in Table 1.
>
> ---
> **A comparison should be made with other methods that utilize retrieval hard prompts/examples for in-context learning.**
>
> Thank you for your suggestion to add more prompt engineering & ICL baselines. We have conducted additional experiments to compare our method with 3 different hard prompt optimization techniques including (1) RoHPr which utilizes the same search technique as RoSPr but retrieves 4 corresponding training instances for in-context learning instead of the corresponding soft prompt, and (2) [APE (Zhou et al., ICLR 2023)](https://arxiv.org/abs/2211.01910) which is an automatic prompt engineering method that utilizes the generation results of an LLM (InstructGPT), known to show better performance than human instructions, and (3) [ZPS (Liao et al., 2022)](https://arxiv.org/abs/2211.04668) which selects an optimal hard prompt from a prompt pool using prompt ensemble and pseudo-labels.
>
>
> As shown in the below table, on average, RoSPr performs the best on average, showing the benefits of using soft prompts over hard prompts for task generalization. While RoSPr shows improvement on 10 out of 11 datasets, other hard prompt optimization methods do not show consistent improvements. We will include these additional results in the revised version.
>
> |       |  RTE  |  CB   | ANLI R1 | ANLI R2 | ANLI R3 |  COPA | Hellaswag | StoryCloze | Winogrande |  WSC  |  WiC  |  AVG  | # Improved |
> |-------|:-----:|:-----:|:-------:|:-------:|:-------:|:-----:|:---------:|:----------:|:----------:|:-----:|:-----:|:-----:|:----------:|
> | T0    | 64.55 | 45.36 |  33.81  |  33.11  |  33.33  | 75.88 |    26.60   |    84.03   |    50.97   |  65.10 | 50.69 | 51.22 |      -     |
> | RoHPr |  47.10 |   45.00  |  33.53  |   33.30  |  33.09  | 73.38 |   26.65   |    86.93   |    50.51   | 64.42 |   50.00  | 49.45 |      3     |
> | APE   | 68.19 | 48.33 |  35.82  |  35.37  |  34.27  |   68.00  |   25.76   |    78.73   |    50.43   | 58.75 | 50.85 | 50.41 |      6     |
> | ZPS   | 58.48 | 60.71 |   36.60  |   34.40  |  33.33  |   76.00  |   28.49   |    87.39   |    51.78   | 64.42 | 50.63 | 52.93 |      7     |
> | RoSPr | 71.54 | 49.48 |  37.05  |  34.64  |  33.92  | 78.75 |   26.97   |    85.52   |    51.50    | 64.52 | 51.76 | **53.24** |     10     |
>
>
> ---
> **Fine-tuning the backbone model to generate soft prompt embeddings poses a challenge for large-sized language models.**
>
> Although fine-tuning the same backbone model is needed to generate soft prompt embeddings, we think that the advantages outweigh the disadvantages for the following two reasons:
>
> - Recent studies, such as [Med-PaLM (Singhal et al., Nature 2023)](https://www.nature.com/articles/s41586-023-06291-2), have demonstrated that training soft prompts on much larger models, such as the 540B PaLM model, can significantly boost the performance in clinical knowledge tasks, showing the feasibility of training soft prompts for >500B LLMs which implies that RoSPr is also scalable.
> - In terms of practicality, [QLoRA (Dettmers et al., 2023)](https://arxiv.org/abs/2305.14314) has shown that the parameter-efficient fine-tuning of 65B LLM is feasible with a single 40GB GPU via 4-bit precision without performance degradation, allowing efficient gradient updates. Especially, since our approach only requires updating 0.007% of the parameters for 150 steps for each soft prompt, we expect that quantization techniques would enable the fine-tuning of large-sized language models with consumer GPUs.
>
> Considering these aspects, we believe that RoSPr can be also practically effective for larger models.
>
> We hope the authors have provided adequate additional experiments and responses to address the weaknesses identified by the reviewer and kindly ask the reviewer to reconsider their initial assessment of the paper.

---

### Official Review · Reviewer_GnhD · 2023-08-13

**Soundness:** 4

**Excitement:**

3: Ambivalent: It has merits (e.g., it reports state-of-the-art results, the idea is nice), but there are key weaknesses (e.g., it describes incremental work), and it can significantly benefit from another round of revision. However, I won't object to accepting it if my co-reviewers champion it.

**Paper Topic And Main Contributions:**

The paper explores the challenge of enhancing zero-shot performance in instruction-following models without extensive computational overhead. By training soft prompts embeddings for hard prompts of source tasks and creating a vector store, ROSPR retrieves relevant prompt embeddings during inference to aid predictions on target tasks. The experiment results highlight the proposed method's efficacy in improving the zero-shot capabilities. Supplementary insights are presented in the analytical section.

**Reasons To Accept:**

The paper provides a well-structured and clear explanation of the proposed framework. The proposed ROSPR method, as evidenced by experimental results, consistently outperforms its baselines on a variety of tasks, only with the inclusion of a relatively small number of parameters. Furthermore, the paper includes a comprehensive ablation study which sheds light on the intricacies of the method.

**Reasons To Reject:**

I find no reason to reject this paper.

**Reproducibility:**

4: Could mostly reproduce the results, but there may be some variation because of sample variance or minor variations in their interpretation of the protocol or method.

**Reviewer Confidence:**

2: Willing to defend my evaluation, but it is fairly likely that I missed some details, didn't understand some central points, or can't be sure about the novelty of the work.

---

> ### Author Rebuttal · Authors · 2023-08-29
>
> Thank you for making the effort to review our paper. We appreciate your recognition of our key arguments that 1) RoSPr consistently outperforms its baselines while adding a small number of parameters and 2) the paper includes a comprehensive ablation. Let us know if there are some improvements that are needed to increase the excitement of the paper.

---

### Official Review · Reviewer_eWKn · 2023-08-13

**Soundness:** 2

**Excitement:**

2: Mediocre: This paper makes marginal contributions (vs non-contemporaneous work), so I would rather not see it in the conference.

**Paper Topic And Main Contributions:**

This paper addresses the enhancement of zero-shot performance in instruction following models through the incorporation of a soft prompt retrieval approach. The main contribution lies in proposing a method to efficiently improve the model's ability to comprehend and follow instructions without the need for explicit fine-tuning. The use of soft prompt retrieval aims to bridge the gap between traditional fine-tuning and zero-shot learning, thereby contributing to more versatile and adaptable instruction following models.

**Reasons To Accept:**

1、This paper aligns with the growing interest in zero-shot learning, showcasing an approach that not only addresses performance limitations but also contributes to advancing the broader field of zero-shot learning.

2、This paper introduces Retrieval of Soft Prompt (ROSPR), which is easily scalable and requires minimal computation by only adding 0.007% parameters to the main model during inference.

**Reasons To Reject:**

1、The conceptual expression in this paper is unclear. Although instruction-following is essentially a form of prompt, the way it is presented in this paper can easily become entangled with the concepts of contemporary LLMs. For instance, the explanation in line 42-52 is not clear.

2、Prompt embedding refers to the representation of instances within the context of prompts, while the concept of zero-shot entails utilizing instances from the training samples' retrieval to enhance downstream tasks. This method has been discussed in [1], yet the authors neither cite it in the related work nor compare their approach to it.

3、This paper is somewhat incremental in its technical approach, lacking significant innovation and contributions.

[1] Decoupling Knowledge from Memorization: Retrieval-augmented Prompt Learning. NeurIPS 2022


**Reproducibility:**

4: Could mostly reproduce the results, but there may be some variation because of sample variance or minor variations in their interpretation of the protocol or method.

**Reviewer Confidence:**

5: Positive that my evaluation is correct. I read the paper very carefully and I am very familiar with related work.

---

> ### Author Rebuttal · Authors · 2023-08-29
>
> Thank you for taking the time to review our paper. We appreciate your recognition of the strengths of the paper: proposing a method to efficiently improve the model's ability to comprehend and follow instructions without the need for explicit target task fine-tuning, thus contributing to advancing the broader field of zero-shot learning.
> We think that all concerns or questions could be addressed. Our response is provided below.
>
> ---
> **The conceptual expression in this paper is unclear.**
>
> Thank you for your feedback. We think that the concept of instruction-tuning was not clearly delivered. Unlike conventional prompt learning, instruction-tuning fine-tunes a base language model $&theta;$ on **multiple** downstream tasks (multi-task instruction learning) $D$={$D_{1}$, .., $D_{T}$}, and performs inference on **held-out tasks**  $G$={$G_{1}$, .., $G_{R}$} that are unseen during fine-tuning. We ensure that $G_{i}$ &notin; $D$.
>
> Instruction Tuning:
> For each iteration,
> 1. Sample $i$ &in; {1, 2,.., $T$}
> 2. Sample {$x_{ik}$, $y_{ik}$} from $D_{i}$ = ($x_{i1}$, $y_{i1}$), .., ($x_{iN}$, $y_{iN}$)
> 3. $max_{&theta;}$ $P(h_{ij}(y_{ik})|h_{ij}(x_{ik}))$
>
> where $h_{ij}$ denotes applying the $j$th hard prompt of the dataset $D_{i}$.
>
>
> Previous works have shown that scaling the number of fine-tuning datasets ($T$) or scaling the model size ($&theta;$) leads to improved generalization to unseen tasks ($G$). However, since these approaches require fine-tuning the full parameters of the model with increased scale which leads to heavy computation, we explore an efficient alternative to enhance the performance by freezing the parameter of an instruction-tuned model and fine-tuning only the soft prompts for each source task. We will clarify this part by adding a preliminary background subsection in the paper.
>
> ---
> **Comparison with [RetroPrompt (Chen et al., NeurIPS 2022)](https://arxiv.org/abs/2205.14704)**
>
> Thank you for the missing reference. We will make sure to include the citation in our revised version. However, we want to point out some differences between RetroPrompt and our method.
> - Although RetroPrompt also adopts a semi-parametric approach to enhance the few-shot/zero-shot performance of LMs for prompt learning, RetroPrompt retrieves representations from the unlabeled training data of the **target task**, while RoSPr ensures a strict held-out setting for **cross-task generalization** by retrieving from source tasks that belong to the different task type (e.g. retrieving paraphrase task soft prompt for entailment tasks).
> - Also, since the labels of the source tasks are normally different from the answer choice list of the target task for our setting, the KNN approach of RetroPrompt which assumes that the retrieved representation has the same answer choice (label choice) as the target task for probability calculation is unapplicable without converting the task into a synthetic set-up (manually matching the label choices between source & target task).
>
> These two reasons (cross-task generalization & unaligned answer choices) are why it is difficult to make an apple-to-apple comparison between RoSPr and RetroPrompt.
>
> ---
> **The novelty of the technique**
>
> Although the idea of retrieval of soft prompts has been explored in previous works ([SPoT (Vu et al., ACL 2022)](https://arxiv.org/abs/2110.07904), [ATTEMPT (Asai et al., EMNLP 2022)](https://arxiv.org/abs/2205.11961)), our work has technical novelty in the following aspects:
> - Our work is the first to observe the effect of retrieval of soft prompt on the **zero-shot** task generalization setting where the training data of the target task does not exist by applying the retrieval process to an instruction-tuned model and showing better performance than hard prompt optimization techniques (see the additional table below) and previous approaches that require target task fine-tuning (PT and ATTEMPT performance on Table 1 of the paper).
> - Other than the retrieval of soft prompt, we additionally introduce Var (variance-based ranking) and Inter+Var (interpolation between soft prompts searched through variance-based ranking) which improve the mean accuracy and reduce prompt sensitivity (STD) of RoSPr shown in Table 1.
>
>
>
> Additional Table 1: Comparison with different hard prompt optimization methods:
> - RoHPr: While the search process is the same as RoSPr, RoHPr retrieves 4 in-context examples (hard prompts) instead of soft prompts.
> - [APE (Zhou et al., ICLR 2023)](https://arxiv.org/abs/2211.01910): Automatic prompt engineering technique utilizing the generation results of an LLM (InstructGPT) that is known to show better performance than human instructions.
> - [ZPS (Liao et al., 2022)](https://arxiv.org/abs/2211.04668): Prompt selection method that chooses the optimal prompt from a prompt pool using prompt ensemble and pseudo-labels.
>
> |       |  RTE  |  CB   | ANLI R1 | ANLI R2 | ANLI R3 |  COPA | Hellaswag | StoryCloze | Winogrande |  WSC  |  WiC  |  AVG  | # Improved |
> |-------|:-----:|:-----:|:-------:|:-------:|:-------:|:-----:|:---------:|:----------:|:----------:|:-----:|:-----:|:-----:|:----------:|
> | T0    | 64.55 | 45.36 |  33.81  |  33.11  |  33.33  | 75.88 |    26.60   |    84.03   |    50.97   |  65.10 | 50.69 | 51.22 |      -     |
> | RoHPr |  47.10 |   45.00  |  33.53  |   33.3  |  33.09  | 73.38 |   26.65   |    86.93   |    50.51   | 64.42 |   50.00  | 49.45 |      3     |
> | APE   | 68.19 | 48.33 |  35.82  |  35.37  |  34.27  |   68.00  |   25.76   |    78.73   |    50.43   | 58.75 | 50.85 | 50.41 |      6     |
> | ZPS   | 58.48 | 60.71 |   36.60  |   34.40  |  33.33  |   76.00  |   28.49   |    87.39   |    51.78   | 64.42 | 50.63 | 52.93 |      7     |
> | RoSPr | 71.54 | 49.48 |  37.05  |  34.64  |  33.92  | 78.75 |   26.97   |    85.52   |    51.50    | 64.52 | 51.76 | **53.24** |     10     |
>
>
> Let us know if there are further questions.

---

### Official Review · Reviewer_YaNx · 2023-08-15

**Soundness:** 3

**Excitement:**

3: Ambivalent: It has merits (e.g., it reports state-of-the-art results, the idea is nice), but there are key weaknesses (e.g., it describes incremental work), and it can significantly benefit from another round of revision. However, I won't object to accepting it if my co-reviewers champion it.

**Paper Topic And Main Contributions:**

The paper under consideration focuses on the enhancement of zero-shot task generalization through retrieval-augmented soft prompting. The authors introduce ROSPR, a method that initiates by storing soft prompt embeddings, followed by their retrieval using a dense retriever. The retrieved soft prompts are subsequently combined based on the Freq./Var. criteria. Across a spectrum of 11 datasets, ROSPR consistently achieves state-of-the-art performance, as substantiated by comprehensive benchmarking. Detailed analysis and ablations are conducted.

**Questions For The Authors:**

* Could you please specify the number of queries employed per test data instance when reporting the results in Table 1? If this count exceeds one, it prompts the question of whether a fair comparison with the baselines is maintained.
* Is there a possibility to adapt the T0-3B soft prompt library to accommodate the T0-11B scenario? Understanding the adaptability of the approach across different library sizes is essential.
* Could you provide insights into the time constraints, with respect to a specific hardware setup, associated with the retrieval of relevant soft prompts from the library? This information would contribute to a more comprehensive understanding of the method's practicality.

**Reasons To Accept:**

* The utilization of a retriever-based source prompt library demonstrates a straightforward and effective approach, leading to the SOTA performance.
* The concept of an evolving source prompt library holds the potential for continuous enhancement of zero-shot performance, capitalizing on the expanding library size.

**Reasons To Reject:**

* The presentation of the paper is weak. In particular, Section 3 lacks clarity and should provide preliminary information rather than solely referencing external sources. Sections 5 and 6 suffer from verbosity and should be revised to more effectively convey findings.
* Figures 3 and 14 lack necessary comparisons with baselines (PT and ATTEMPT), which is crucial for contextualizing the contributions.

**Reproducibility:**

4: Could mostly reproduce the results, but there may be some variation because of sample variance or minor variations in their interpretation of the protocol or method.

**Reviewer Confidence:**

3: Pretty sure, but there's a chance I missed something. Although I have a good feel for this area in general, I did not carefully check the paper's details, e.g., the math, experimental design, or novelty.

---

> ### Author Rebuttal · Authors · 2023-08-29
>
> Thank you for taking the time to review our paper. We appreciate your recognition of our key arguments that 1) a retriever-based source prompt library demonstrates a straightforward and effective approach and 2) an evolving source prompt library holds the potential for continuous enhancement of the performance.
>
> We think that all concerns or questions could be addressed.
>
> First, we provide the response to the 2 Reasons To Reject.
>
> ---
> **The presentation of the paper is weak.**
>
> We agree that adding a preliminary background subsection introducing the conventional instruction-tuning process and explaining how RoSPr could be applied to instruction-tuned models would clarify the process of RoSPr. We will add the corresponding subsection to the paper. Also, for Sections 5 and 6, we will reduce redundancy and verbosity to effectively convey our results and findings in our revised version.
>
> ---
> **Figures 3 and 14 lack comparisons with PT and ATTEMPT.**
>
> Thank you for your suggestion. For Figure 3, since the official Big-bench is a zero-shot benchmark that does not provide any training datasets, it is inapplicable to measure the performance of PT and ATTEMPT since both methods require target task training datasets. This again indicates the wide applicability of RoSPr, since it could be applied to target tasks where the training data does not exist, unlike PT and ATTEMPT.
>
> For Figure 14, we provide the (Min, Median, Max) values for PT, ATTEMPT, and RoSPr. We will also add the results in the paper.
>
> |         |          RTE          |          CB          |        ANLI R1        |        ANLI R2        |        ANLI R3        |          COPA         |       Hellaswag       |       Winogrande      |          WSC          |          WiC          |
> |---------|:---------------------:|:--------------------:|:---------------------:|:---------------------:|:---------------------:|:---------------------:|:---------------------:|:---------------------:|:---------------------:|:---------------------:|
> | PT      | 57.76 / 63.54 / 67.87 | 8.93 / 48.21 / 58.93 | 32.10 / 33.00 / 34.40 | 30.10 / 31.00 / 33.40 | 31.83 / 33.08 / 34.08 | 64.00 / 74.50 / 78.00 | 25.56 / 26.59 / 27.20 | 49.33 / 50.59 / 52.49 | 36.54 / 48.08 / 53.85 | 49.84 / 50.00 / 50.31 |
> | ATTEMPT | 65.70 / 71.03 / 73.65 | 8.93 / 50.00 / 62.50 | 33.30 / 36.70 / 39.00 | 32.67 / 34.58 / 36.50 | 32.67 / 34.58 / 36.50 | 64.00 / 76.00 / 81.00 | 25.92 / 26.65 / 28.16 | 51.07 / 51.22 / 51.70 | 63.46 / 63.46 / 68.27 | 49.06 / 49.92 / 56.74 |
> | RoSPr   | 58.12 / 72.57 / 76.53 | 8.93 / 50.00 / 67.86 | 33.30 / 38.10 / 40.50 | 33.30 / 34.90 / 36.00 | 32.42 / 33.75 / 35.58 | 67.00 / 80.50 / 83.00 | 26.58 / 26.96 / 27.70 | 50.59 / 51.38 / 53.59 | 59.62 / 65.38 / 68.27 | 50.00 / 51.88 / 53.76 |
>
>
> Next, we address the 3 Questions for the Authors.
>
> ---
> **Q1: The number of queries employed per test data instance.**
>
> Thank you for your question. Note that the queries employed during inference for retrieval are not per test data instance but per **test dataset**. Following [ReCross (Lin et al., NeurIPS 2022)](https://arxiv.org/abs/2204.07937), we sample 32 test instances randomly and search for the optimal soft prompt at the beginning of the zero-shot adaptation of each dataset. This approach does not require a ground truth label for the queries employed or a separate validation set for each test dataset but leads to better performance than fine-tuning baselines (PT and ATTEMPT) which require a labeled training set. Also, instead of searching for the optimal soft prompt for each single query, the retrieval process only occurs once for a test dataset for this approach, reducing the retrieval latency.
>
> ---
> **Q2: Adapting the T0-3B soft prompt library to accommodate the T0-11B scenario**
>
> Although we have not tested on the T0-11B scenario due to computational limitations, the soft prompt library could be adapted to the T0-11B scenario as well either by 1) training soft prompts of T0-11B through source prompt tuning, or 2) learning a linear layer to adapt the prompt library constructed from T0-3B to the dimension size of T0-11B embeddings. We leave adaptation to larger language models as future work.
>
> ---
> **Q3: Time constraints of the retrieval of relevant soft prompts from the library.**
>
> Thank you for your question. For a single A6000 GPU, our dense retrieval process for each target task takes 2.07 seconds on average. We choose a 35M-sized dense retrieval model (T5-small encoder) to reduce the latency during inference, which is much smaller than other retrievers such as [ReCross (Lin et al., NeurIPS 2022)](https://arxiv.org/abs/2204.07937) (400M) or [Setence-bert (Reimers & Gurevych, EMNLP 2019)](https://arxiv.org/abs/1908.10084) (80M). Also, we only store 100 examples per training prompt for efficient search.

---

### Meta-Review · Area_Chair_LPEM · 2023-09-18

**Recommendation:** 3

**Metareview:**

This work effectively leverages soft prompt retrieval to improve the zero-shot performance of instruction-following models and provides an easy method to improve instruction-following capabilities without fine-tuning. The reviewers raised some concerns about unclear writing and we encourage the authors to improve the presentation.

---

### Decision · Program_Chairs · 2023-10-07

**Decision:**

Accept-Findings

**Comment:**

This work effectively leverages soft prompt retrieval to improve the zero-shot performance of instruction-following models and provides an easy method to improve instruction-following capabilities without fine-tuning. The reviewers raised some concerns about unclear writing and we encourage the authors to improve the presentation.